# Why Do Companies Choose Female CEOs?

**Shuo Han** [1] 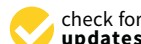**, Weijun Cui** [1,2,*]**, Jin Chen** [3,4] **and Yu Fu** [1]

[1] Business School, Nanjing University of Information Science & Technology, Nanjing 210044, China
[2] China Institute of Manufacturing Development, Nanjing University of Information Science & Technology, Nanjing 210044, China
[3] School of Economics and Management, Tsinghua University, Beijing 100085, China
[4] Research Center for Technological Innovation, Tsinghua University, Beijing 100085, China
[*] Correspondence: cuiweijun@nuist.edu.cn

**Abstract:** For the career development of chief executive officers (CEOs), the existing literature lacks research from the perspectives of gender and the environment. Starting with the perspective of the environment, and through the adoption of the World Bank Survey Data on Chinese Enterprises and China's Comprehensive Social Survey (CGSS), this paper addresses the question: "Why do companies choose female CEOs?" The analysis examines 15 aspects, including enterprise scale, age, industry, owner, product market, market environment, city level, etc. The research findings are as follows. (1) The corporate environment has an impact on CEO gender, and there are differences in its nature and the degree of impact. Enterprise size, state-owned shareholding, radiation effect, market environment, and gender culture have a significant negative impact on female CEOs, while product features, industry characteristics, and female owners have a significantly positive impact on female CEOs. (2) In terms of the impact mechanism, the impact of the meso-environment on female CEOs is significantly adjusted by the macro-environment. This paper extends the existing theory from the perspectives of gender and the environment. Relevant policy recommendations are proposed to provide a theoretical basis for the government to promote women's employment policies and provide effective suggestions for promoting women's career development.

**Keywords:** female CEOs; environment; regulating effect; gender culture

## 1. Introduction

Women are an important force in achieving sustainable development, and gender balance is very important for enterprise development. In recent years, with the improvement of women's education levels, social and economic development, and the progress of science and technology, the proportion and number of female executives are also increasing. This phenomenon has also aroused widespread concern in academia [1]. However, as we have observed, the position of corporate chief executive officers (CEO) is still dominated by males and rarely held by women.

As the existing theories of career development are concerned, neither occupational status attainment theory nor human resource theory can reasonably explain why there are fewer female CEOs. Both theories emphasize the importance of personal factors to career development. According to the occupational status attainment theory, with economic growth and rising education levels, people's career development will depend on individual efforts, namely, their achievement factor, rather than such ascribed factors as family background, etc. Human capital theory holds that education, work experience, and skills are all rare resources of economic value and of great importance to career development. However, as we see, with women's education levels going up and their years of working and working experience increasing, their positions and levels have not improved accordingly. Only

a few women have made their way to CEO, breaking through the glass ceiling. Therefore, the few women who break through the glass ceiling and become CEOs are more worthy of our research.

So, from what perspective should we look at and understand the unusual behavior of companies employing female CEOs? Theoretically, the drawback of the existing theories consists of the ignorance of the impact of the company's environment. The company's workplace revolves around men, and as a result, women have to learn through socialization and adapt to the environment to meet the demand of enterprise development. Therefore, the environment is the key to understanding companies recruiting female CEOs and breaking through the existing theories. In summary, we should study the gender of CEOs from the perspective of corporate environment, namely: how does the environment affect the gender of CEOs? The answer to this question will help us better understand and evaluate female CEOs and promote sustainable enterprise development.

In view of the factors affecting CEO genders, scholars have carried out research regarding women's own features [1], company characteristics [2], industry environment [3], cultural differences [4], and other different perspectives, and have drawn many useful conclusions. These studies have both strengths and shortcomings. First of all, there is a hypothesis behind previous studies that males have stronger management ability than their female counterparts. However, gender is a kind of social construction, and gender culture has a profound impact on the viewpoints toward males and females. In fact, as Simone de Beauvoir, pioneer of the world feminist movement, put it like this, "Women are not born, but acquired" [5]. Thus, the stereotyped cognition of women may more or less affect existing research. Secondly, existing research studies lack an overall analysis framework, and the mechanism analysis of the impacts of each factor is insufficient. Although different impact factors feature in in-depth research studies, few studies in the literature have put different factors into a framework. In fact, the designation of CEOs is a strategic issue that may be affected by the environment. However, little of the existing literature has analyzed it from the perspective of the environment. In summary, existing research studies rarely provide clear and credible answers to this issue [6].

Based on the above analysis, the core viewpoint of this paper is that there are no differences between male and female CEOs in terms of ability, and the choice of female CEOs is a result that may be affected by the corporate environment. The purpose of this paper is, on the basis of relaxing the original hypothesis and considering that there are no differences between male and female CEOs in terms of ability, to construct an overall framework of the corporate micro-environment, industry meso-environment, and regional macro-environment from the perspective of the environment, and conduct studies that ask: "why do companies choose female CEOs?" In response, this paper mainly answers from two aspects. (1) What environmental factors have a significant impact on female CEOs? (2) What is the impact mechanism of the environment on CEO gender?

The main approach adopted in this paper is to start with the corporate environment and conduct research by selecting the World Bank Survey Data on Chinese Enterprises 2010 and China's Comprehensive Social Survey (CGSS 2012). To be specific, the corporate micro-environment is described in relation to enterprise size, enterprise age, company attributes, industry characteristics, female owners, and other aspects. The industry meso-environment in which enterprises are located is described in relation to competition order, market environment, and other aspects. The regional macro-environment in which enterprises are located is described from an urban economic level, radiation effect, city level, gender culture, and other aspects. In short, the impact of the corporate environment on CEO genders and its mechanisms have been studied.

The research findings are as follows. (1) The environment in which enterprises are located has a significant impact on CEO genders. In the corporate micro-environment, compared with other factors, enterprise size, state-owned shareholding, product features, industry characteristics, and female owners have significant impacts on female CEOs. In the industry meso-environment, the market environment is more important, yet it has a significantly negative impact on female CEOs. In the macro-environment, the radiation effect and gender culture have significantly impacts on female CEOs. (2) As far as the mechanism of action is concerned, CEO genders are by no means

impacted between various environments separately, but there is a kind of regulating effect. The gender culture in the macro-environment has a positive regulating effect on the market environment in the industry environment.

The main contributions of this paper comprise the following. (1) The existing human resource theory has been supplemented in a positive way. The existing human resource theory and occupational status attainment theory both stress the individual's own value and its realization, but ignore the external environment, which is instead the key to explaining and understanding a company's employment of female CEOs. Therefore, from the perspective of gender and the environment, this essay makes up for the shortcomings of the existing research and theories, and expands the existing theoretical framework. (2) In terms of research design, this paper argues that there are no differences between males and females in business management ability, which relaxes the original research hypothesis, and provides a complete environment analysis framework for analysis of female executives, especially female CEOs, from an environmental perspective, and clarifies the impact mechanism between environments. This paper argues that gender is the result of social construction, and it is impacted by the gender culture. Thus, there are no differences in ability between males and females. Meanwhile, corporate characteristics and the environment serve as the key factors to explain how women break through the ceiling and enter into the top management of businesses [7]. Therefore, this paper provides a complete environment analysis framework, further clarifies the impact mechanism of different environments on CEO genders, and enriches existing research studies. (3) This paper is conducive to a reasonable view, understanding, and evaluation of female CEOs, thus providing a reference for women's career development. The results of this paper show that female CEOs are impacted by gender culture, industry differences, corporate characteristics, and other aspects. Females' assumption of the office of CEO is the result of the environment. Thus, there is need to treat female CEOs more fairly and rationally, so as to provide a good development space for female executives' career development and also provide reference for women's career development. (4) This paper also supplies an effective reference experience for countries with economies in transition through the use of Chinese data. From the perspective of sample selection, there are few existing studies on China. The Chinese economy is in a critical period of transformation and development, and its market mechanism is not yet sound, thus providing a broad space for exploring new concepts, theories, and insights [8]. Combined with the economic background of China's transition period, this paper provides an effective reference experience for countries with transition economies or countries with imperfect market mechanisms with regard to the increase of the proportion of female CEOs and the exertion of female CEOs' abilities, and also provides more empirical evidence for deepening the cognition of female CEOs.

The main structure of this paper is as follows. Firstly, the relevant literature is reviewed from the micro, meso, and macro perspectives, and shortcomings in existing research studies are analyzed. Secondly, a relatively complete analysis framework is constructed from the perspective of the enterprise environment, and empirical analysis is conducted to test the robustness of the regression results. Finally, the research conclusions are summarized, and policy recommendations are proposed.

## 2. Literature Review

### 2.1. Sex and Gender Socialization

Sex is an objective fact. With the development of productivity, gender culture was gradually formed. It is well-known that sex is caused by physiological differences, which is an objective fact that cannot be changed and must be recognized. In the primitive society, the basic needs of existence formed a desire for reproduction. Due to the ignorance of reproductive science, since women can bear children, respect and worship for women was formed in primitive societies. With the improvement of productivity level and the gradual formation of the social division of labor, men gradually mastered the dominant positions in production and life due to their innate advantages. After that, family and

marriage patterns also underwent major changes, and this change was also gradually strengthened, thus forming the male-dominated gender culture [9].

Under this gender culture, people gradually formed their understandings and even prejudices against male and female. Therefore, gender is the inevitable outcome of the socialization process of sex. The essence of gender is social construction. Gender culture is a basic component of human society and culture, and it had a major impact on people's productivity and lives [10]. Gender culture refers to the values, ethics, knowledge experience, customs, system norms, and other ideologies and their manifestations developed on the basis of social characteristics, social behaviors, and social relations between males and females [10]. Its core is to divide human beings into males and females, assign them different roles with separate connotations, identify different cultural instructions, and standardize different behavioral logic and development paths. Thus, the essence of gender is social construction, and it is influenced by gender culture [11].

## 2.2. Female and Sustainable Development

In terms of the relationship between women and sustainable development, women and sustainable development promote each other and influence each other, as shown in Figure 1. On the one hand, promoting women's development and gender equality are inevitable choices for sustainable development. Sustainable development is inseparable from women's participation. The level of development of women will not only affect the coordinated development of society, it will also affect the way in which generations can achieve sustainable development. On the other hand, sustainable development further protects the legitimate rights and interests of women. Women will also develop themselves in the process of participating in sustainable development.

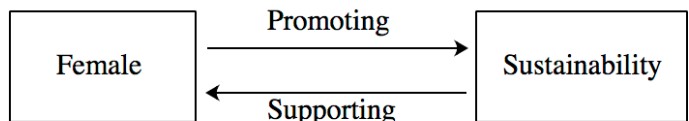

**Figure 1.** Relationship between women and sustainable development.

So, how can women promote sustainable development? As shown in Figure 2, first of all, in politics, respecting and protecting women's interests is one of the standards of social progress. Women's right to education and protection needs to be further strengthened. In terms of goals, gender equality is an inevitable option for achieving sustainable development. In September 2015, the United Nations Conference on Sustainable Development was held and officially approved, which was called *Transforming our World: The 2030 Agenda for Sustainable Development* and proposed the goal of "realizing gender equality and empowering all women and girls". Today, gender discrimination is prevalent, as the gap between women and men remains in employment, medical care, and political rights, etc. The United Nations statistics show that as of 2014, 143 countries have guaranteed gender equity in their constitutions, but 52 countries have yet to take the step.

Second, in terms of economics, female CEOs can reduce corporate risk, promote corporate innovation, increase corporate value, and promote sustainable economic development. At the enterprise level, the executive ladder theory [12,13] holds that the heterogeneity of the executive's characteristics affects corporate policies, risks, and performances [14]. In light of the relationship between women and a company's sustainable development, on the one hand, women constitute an important part of corporate governance from the inside and increasing the number of women on the board of directors raises the value of enterprises [15]. Female CEOs have improved the performance of businesses [2]. In companies under complex environments, when the proportion of female executives is high, it can indeed produce positive and significant returns [16] compared to other companies with fewer female executives. At the same time, female CEOs promote corporate technological innovation more than male CEOs [9]. On the other hand, as for the outside the company, the increase in the number of female executives improves the quality of corporate information disclosure [17]. Therefore, the

female CEO is conducive to improving the enterprise's value, promoting its innovation, and ensuring its sustainable development.

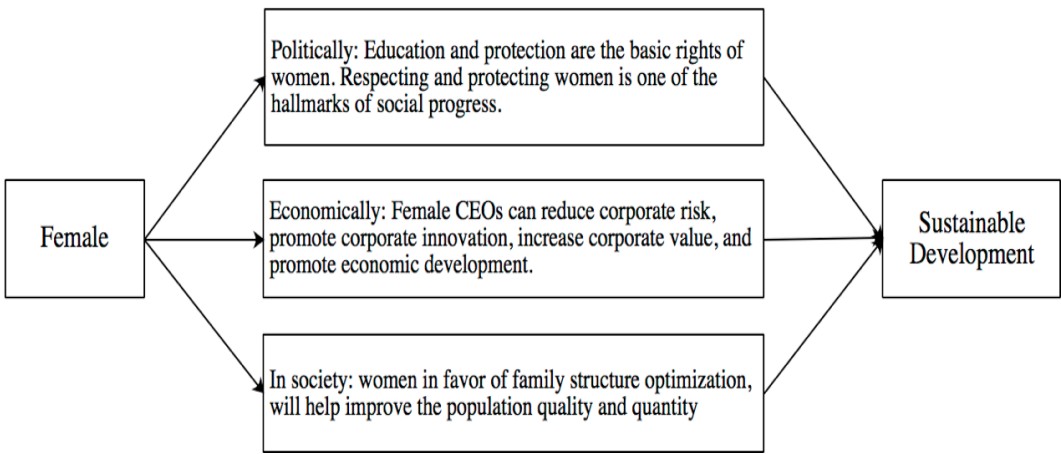

**Figure 2.** Mechanism for women to promote sustainable development.

Third, from a social perspective, women can promote the optimization of family structures and improve the quality and quantity of the population. Sustainable development requires the broad participation of the whole society in light of the driving force, and women are an important force in achieving sustainable development. Women and girls make up half of the world's population and therefore takes up half of the world's potential. As far as sustainable development is concerned, the relationship between population quantity and quality needs to be coordinated. In light of physiological function, women are essential to sustainable development, as population growth is the inexhaustible driving force. At the same time, in terms of population quality, women play a crucial role in the education and bringing up the next generation. A lack of maternal love affects the healthy growth of children.

In summary, women are an important driving force for the sustainable development of the whole society, to which the participation of women is more favorable. Therefore, it is necessary to give full play to the role of women in the sustainable development of the whole society. At the same time, corporate governance requires a new sustainable governance model that effectively unleashes the combined potential of men and women in the enterprise [18]. In this model, the CEO is at the core, and thus CEO gender is of great significance in corporate sustainability.

*2.3. Environment and CEO Gender*

2.3.1. Definition and Composition of the Environment

The environment has a major influence on enterprise operation and management. The environment provides conditions for the survival and development of enterprises. From the perspective of system theory, as an open system, an enterprise is a subsystem that is subordinate to the industry, society, country, and even the whole world. In enterprise management, a correct understanding and analysis of the environment is a prerequisite for the correct formulation of the strategy. In management research, the environment has an imperceptible but considerable influence on organizational behavior [19].

For a company, the position of CEO is very important, and generally will not be easily replaced. This means that the selection of CEO is a major strategic decision that affects the development of the company. During the process of research on female executives, corporate characteristics play a key role in accounting for women's entry into management [7]. At the same time, the selection and appointment of senior executives is a typical strategic behavior and organizational result that is subject to the level of industry and social and economic development. Therefore, building an integrated environment is very helpful for studying the gender of CEOs.

We believe that the environment is a collection of factors that influence and constrain the production and operation activities of enterprises. The environment includes not only the individual micro-environment, but also the macro-environment such as the country, and the meso-environment between individuals and countries [20]. Therefore, according to the sphere of influence of the environment, the environment can be divided into micro factors, meso factors, and macro factors [19,20]. In this essay, the micro-environment mainly refers to the internal characteristics of the enterprise, including the age and scale of the enterprise, shareholding structure, the scope of sales of the products, etc. The meso-environment refers to the characteristics of the sector and industry, including the market environment and competition order. The macro-environment mainly includes the characteristics of a country or city, specifically containing economic level, political level, cultural environment, etc., as shown in Figure 3.

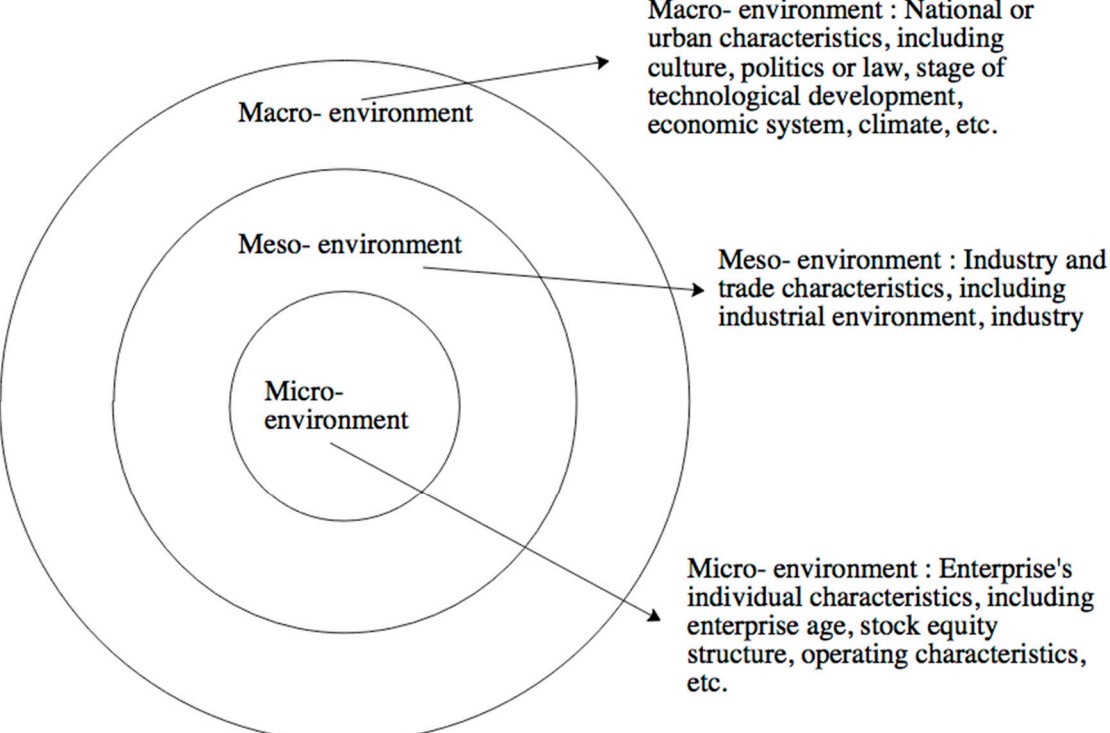

**Figure 3.** Classification of Environment.

2.3.2. The Importance of the Influence of the Environment on the Gender of the CEO

As shown in Figure 4, the environment has a major impact on the gender of the CEO. First, the environment calls for new requirements on enterprise development. Changes in the environment have forced enterprises to adjust their business strategies to meet the requirements of their development, and those that actively respond to changes in the external environment are the ultimate successful enterprises. Therefore, the environment can impose new requirements on CEO abilities, in which only those who meet the new requirements can become CEOs. Second, the environment is the implementer of women's socialization. In the process of women's socialization, families, schools, work units, newspapers, books, movies, and networks are all providers of women's socialization, and they are all part of the environment.

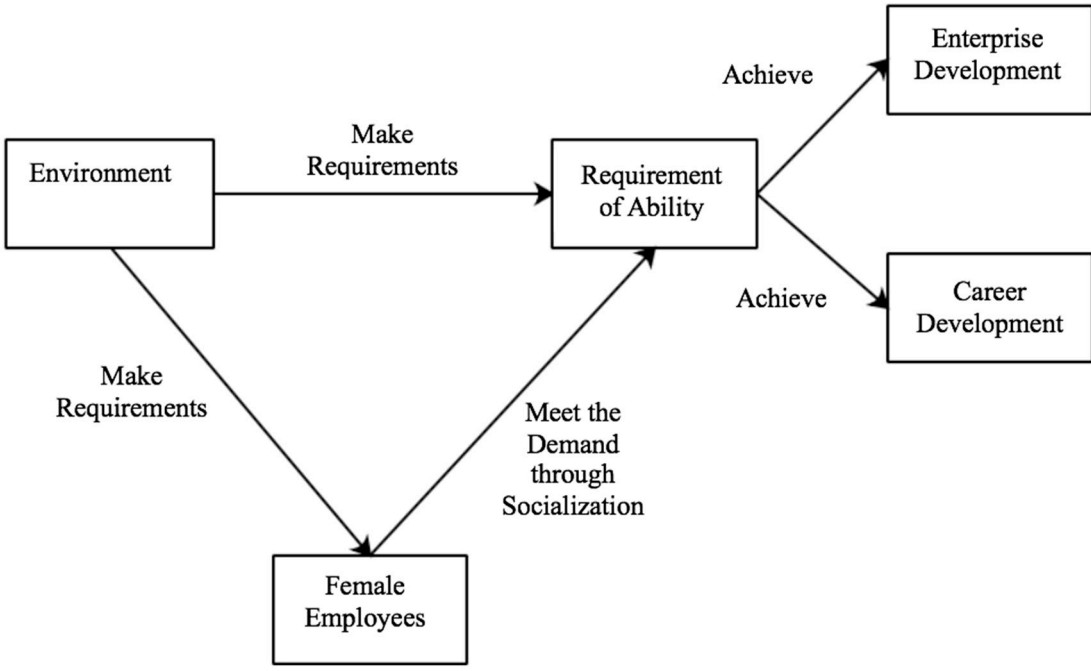

**Figure 4.** Illustration of the impact of the environment.

Therefore, the environment influences women to achieve socialization through learning, interaction, and education. Only those women who meet the requirements of the environment can meet the requirements of enterprise development and become CEOs.

### 2.3.3. Impact of Environment on CEO Gender

At the micro level, the existing research mainly involves the impact of enterprise internal characteristics on the CEO's gender, including enterprise age, performance, scale, and corporate investors. As regards corporate performance and risk, it was discovered that white women and women of color are more likely to be appointed as CEOs by companies with poor performance, while white men may replace these white women and women of color when business risk increases and performance declines [21]. In terms of corporate size, the academic community has not yet reached a consensus, which is probably because there are differences in the measurement of corporate size [22]. Some researchers believe that bigger companies are more favorable toward female CEOs [23], while others have noted that smaller companies are more likely to employ female CEOs [23]. In terms of company age, Gul, Srinidhi, and Ng found that among the United States (U.S.) listed companies, the older the company, the more female senior managers are employed [23]. However, Morikawa found that among the listed companies in Japan, the younger a company is, the more likely it is to choose female senior managers [24]. With regard to exterior perspectives, Brinkhuis and Scholtens researched from the perspective of investors and found that investors have no impact on which gender should be chosen for the position of a CEO [25]. Female CEOs and female-biased recruitment strategies are significantly more favorable for women to become managers. However, these studies also found that corporate internationalization and foreign investment have a negative impact on female senior managers [7]. Moreover, the subsidiaries of parent companies and companies with trade unions have a negative impact on female senior managers [24].

At the meso level, the existing research mainly focuses on the impacts of industry on the CEO's gender, yet the impacts from different industries on the CEO's gender are different. In the United States, there are more female senior managers in service industries than in other industries, especially in health and social services [26]. Other studies have also revealed that there are more female directors in retail sectors and fewer female directors in STEM (STEM is the abbreviation of the initials of science,

technology, engineering and mathematics.) and financial sectors [27]. From the market point of view, a sound market system can make a comprehensive assessment of the company's development. However, existing studies show that financial markets cannot accurately evaluate the tenure performance of female CEOs [28], the stock market does not pay enough attention to the turnover of female CEOs, and the negative market reflection is small [3].

At the macro level, the existing research mainly focuses on the impacts of the culture, system, and political system on the CEO's gender. With respect to culture, the degree of cultural equality alleviates the differences in career development between women and men [29], and religion does not affect the status of women on the board of directors [4]. Countries with less emphasis on family values have more women directors [30,31]. As regards institution, transnational differences in institutional structures and gender systems have affected women's chances of entry onto boards of directors [32]. At the same time, Grosvold, Rayton, and Brammer chose 23 countries as samples and found that family, education, economy, and government have significant positive impacts on women's entries onto boards of directors [29]. In terms of political systems, it is found that countries with a long tradition of women's political representations are unlikely to have a high level of female board representatives [33].

In summary, the existing research is not complete in the following aspects.

First, they did not lay particular emphasis on the CEO's gender as much as they focused on the number and proportion of female senior managers. There are few in-depth studies on the quality of employment of female senior managers. However, as women gradually become a major force of social and economic development, attention should be paid to not only the number and proportion of female senior managers, but also the quality of their employment. Only when the ratio of CEO genders is improved can talents be fully utilized. In general, companies will choose male CEOs, but few women will be CEOs. So, what are the reasons for choosing women as CEOs compared to men? Existing research has failed to give a clear answer.

Second, they did not have a systematic analytical framework, and the analysis of the influence mechanism of each environment was not sufficient. Although in-depth studies have been conducted in micro, meso, and macro environments, few studies have placed micro, meso, and macro environments into one framework. The process of recruiting senior managers, and CEOs in particular, may be influenced by the corporate-level micro environment, the industry-level meso environment, and the regional-level macro environment. Thus, it is quite necessary to place micro, meso, and macro environments into one framework for analysis purposes.

Third, from the perspective of sample selection, existing studies are mainly concentrated in countries and regions such as Europe and the United States, yet few studies have been conducted on China. As the world's second largest economy and a non-immigrant country, China has a different cultural tradition and connotation compared with that of the Western world. The particularities of China are mainly reflected in the following aspects. (1) Economically, China is now at the critical period of transformation and development. In view of China's large economic volume and difficulty in transformation, it is important to pay attention to the processes required to efficiently utilize feminine human power capital. (2) There is an imbalance in China's economy; that is, there is a relatively large regional gap in China's economy, which provides a good test ground for studies of economic differences and female senior managers. (3) Culturally, China is a unified and multi-ethnic country, rather than a non-immigrant country, thus presenting the characteristics of multiculturalism featured with great cultural differences within the region and attitudes toward women. Therefore, in the context of transformation and development, China's economic imbalance and cultural diversity are of great significance to the studies conducted on the quality of employment of female senior managers during the transition period.

Based on the above analysis, this paper constructs an overall framework featured with a micro environment that is mainly represented by corporate characteristics, a meso environment that mainly includes the industrial environment, and a macro environment that mainly involves regional economy, gender culture, and so on. This paper analyzes the impacts of the corporate-level micro environment,

industry-level meso environment, and regional-level macro environment on the quality of employment of female senior managers in the Chinese context, supplementing the insufficiencies in existing research studies.

## 3. Research Design

### 3.1. Data Source

The empirical data in this paper are derived from the World Bank's Questionnaire Survey on Chinese Enterprises 2012 and CGSS 2012, *Report on China's Provincial Marketization Index 2016*, Google Maps, and Sample City Statistics Yearbook

The collection of World Bank Survey Data on Chinese Enterprises 2012 was conducted from December 2011 to February 2013, and the World Bank adopted the stratified sampling method in China to investigate 2848 Enterprises across 23 cities (the provinces and cities mainly involved in the World Bank's Survey include Beijing, Shanghai, Guangdong: Guangzhou, Shenzhen, Foshan, Dongguan, Anhui: Hefei, Hebei: Shijiazhuang, Tangshan, Henan Province: Zhengzhou City, Luoyang City, Hubei Province: Wuhan City, Jiangsu Province: Nanjing City, Wuxi City, Suzhou City, Nantong City, Liaoning Province: Shenyang City, Dalian City, Shandong Province: Jinan City, Qingdao City, Yantai City Sichuan Province: Chengdu, Zhejiang Province: Hangzhou, Ningbo, and Wenzhou) and within 12 provinces (i.e., municipalities), including Beijing, Shanghai, and Guangdong province. Among them, there were 2700 private enterprises and 148 state-owned enterprises. The survey provided the data and additional information corresponding to one complete fiscal year of the chosen enterprises. The reasons for choosing the data from the World Bank's Questionnaire Survey on Chinese Enterprises 2012 were based on the following three aspects. First, due to the availability of data, micro-data at the enterprise level is difficult to obtain publicly in China, and such data is the latest data from the World Bank Survey on Chinese Enterprises that is publicly available at present. Second, in terms of sample selection, the reason why the data of listed companies was not selected in this paper is because non-listed companies constitute the main body of the Chinese market. As of 2016, there were 3337 enterprises in the Shanghai and Shenzhen Stock Exchanges, whereas there were 87.054 million market entities of various kinds in China (statistics on the Number of Private Enterprises in China, www.chinabgao.com, http://www.chinabgao.com/k/qiye/37764.html). Thus, the proportion of listed companies in China remains relatively low, while the market entities are non-listed companies. Third, compared with the non-listed companies, the Chinese listed companies have better corporate governance characteristics and a more advanced system. Therefore, based on the comprehensive consideration of the market and enterprise entities and the situation of corporate governance, the data from the World Bank's Questionnaire Survey on Chinese Enterprises 2012 can reflect the real situation of Chinese enterprises in a more objective manner.

To ensure the accuracy of gender culture data, this paper chooses the authoritative and accurate data from CGSS. There are two main reasons for the selection of this data. First, CGSS data can guarantee the authoritativeness. Founded in 2003 and implemented by Renmin University of China, CGSS is the earliest nationwide comprehensive and continuous academic investigation project in China. In accordance with international standards, since 2003, CGSS has conducted continuous cross-sectional surveys of more than 10,000 households throughout provinces, municipalities, and autonomous regions of China every year. CGSS systematically and comprehensively collects data from multiple levels—the society, community, family, and individual—summarizes the trends of social changes, and probes into issues of great scientific and practical significance. Articles based on CGSS data have also been published in more than 1000 academic journals. In 2007, CGSS represented China and became a member of the International Social Survey Program and served as a window of China for international exchanges and cooperation in social surveys. Moreover, CGSS data is considered to be at par with the World Bank's survey data. CGSS provides the data from all provinces, municipalities, and autonomous regions throughout mainland China, including those sample cities surveyed by the World

Bank. However, unfortunately, due to confidentiality requirements, CGSS data only published the province information of samples, while the information about the prefecture-level cities of samples remains unpublished. Therefore, the variables of gender culture in this paper reflect the overall perspective of residents of such province toward women.

The market environment data is derived from the data of 2012 in *Report on China's Provincial Marketization Index* prepared by Fan Gang and Wang Xiaolu. The *Report on China's Provincial Marketization Index* evaluates the overall progress of market-oriented reforms in various provinces, autonomous regions, and municipalities directly under the central government throughout China from 2008 to 2014, and charts progress from different aspects. Such data is widely quoted by both Chinese and foreign scholars.

The center distance comes from Google Maps. We used Google Maps to measure the nearest road distance from the sample cities to Beijing, Shanghai, Guangzhou, and Chongqing, and conducted the normalization processing. The four central cities of Beijing, Shanghai, Guangzhou, and Chongqing were chosen because the planning and positioning of the five national central cities of Beijing, Tianjin, Shanghai, Guangzhou, and Chongqing were clarified in the *National Urban System Planning (2010–2020)* issued by the Ministry of Housing and Urban–Rural Construction in February 2010. As the capital of China, Beijing is about 137 km away from Tianjin. To avoid complicating the research process, this paper did not choose Tianjin as a central city. The reason for choosing highway distance is that people have not yet realized the convenience of high-speed railway due to the limited routes of China's high-speed railways in 2012 and the railway network that is in progress. For this reason, we selected highway distance instead of high-speed railway time.

The economic levels of sample cities are derived from the statistical yearbooks of each region.

*3.2. Sample Selection*

For the sake of efficient research, the raw data from the World Bank were screened and processed in this paper. The main steps are as follows. (1) In consideration of the matching attributes of questionnaire and samples, sample data marked with "No, does not match" are deleted according to Question A5 (Question A5: Sector match between screener information and sample frame; three options are given: 1. Yes, the screener and sample frame info match; 2. No, the screener and sample frame do not match, but the establishment still does activities that match the sample frame; and 3. No, does not match). (2) In consideration of the authenticity and validity of the data, the sample data marked with "Not truthful" and "Are arbitrary and unreliable numbers" are deleted according to questions A16 (Question A16: It is my perception that the responses to the questions regarding opinions and perceptions; three options are given: 1. Truthful; 2. Somewhat truthful; 3. Not truthful) and A17 (Question A17: In responses to the questions regarding figures (productivity and employment numbers), three options are given: 1. Are taken directly from establishment records; 2. Are estimates computed with some precision; and 3. Are arbitrary and unreliable numbers). (3) For the sake of prudence, the missing values were deleted among all the models, and no method was used for replacements.

The treatment processes of CGSS in this paper are as follows. (1) Samples of missing values that did not answer the question were deleted. (2) For question A423 (Question A423 in CGSS: Do you agree with the following statements: A good husband is better than a good job? Four answers are provided for this question: "Completely disagree", "Partially disagree", "It doesn't matter whether I agree or disagree", "Partially agree", and "Completely agree"), reflecting the gender culture, the average value was obtained by provinces, that is, the sample data in the same province was aggregated to obtain the average values by which the degree of gender culture corresponding to such province can be measured.

Other data processing are as follows: (1) market environment data was derived from the data of 2012 in the *Report on China's Provincial Marketization Index*; (2) economic development level is represented by the per capita gross domestic product (GDP) in the statistical yearbook of every region;

and (3) central distance is the highway distance between sample cities and central cities as measured on Google Maps.

### 3.3. Empirical Model

Since this paper mainly analyzes the internal characteristics and external environments of enterprises, no control variables were set in the empirical model. In this paper, the following models are constructed: $y = \alpha_0 + \alpha_1 Micro + \alpha_2 Meso + \alpha_3 Macro + \varepsilon$.

Among them, $y$ denotes the core variable of the CEO's gender, *micro* denotes the corporate-level micro environment, *Meso* denotes the industry-level meso environment, *macro* denotes the macro environment of economy, politics, culture, and so on, $\alpha_0$0 denotes the constant term, and $\varepsilon$ denotes the random error term.

### 3.4. Design and Description of Variables

The specific contents of variable design are reported in Table 1. In this paper, the CEO's gender is taken as the explained variable, while the micro, meso, and macro environment in which an enterprise is involved are taken as the explaining variables. The descriptive statistical results of the samples are reported in Table 2.

#### 3.4.1. Micro-Environment: Company Characteristics

As regards the corporate-level micro-environment, this paper sets variables such as the basic company characteristics, stock equity characteristics, and operating characteristics.

In terms of basic features, this paper refers to the research outcomes of Morikawa [24] and Kirsch [6], and sets variables such as enterprise size, enterprise age, corporate attributes, industry characteristics, stock equity characteristics, and operating characteristics. The enterprise size is set as the logarithm of the number of regular employees. It is very important to measure the development level and capability of an enterprise by its scale, and there are many different ways of setting up the enterprise [22]. However, it is subject to the questionnaire setting and data availability, referring to previous studies [34]. With regard to the enterprise age, one sample out of all the samples was established in 2012, while three samples were established in 2011. Considering that the survey was conducted from 2011 to 2012, the enterprise age of all these three samples was set as 1. Meanwhile, the enterprise age is set to add 1 after the logarithm was taken, for eliminating the effect of heteroscedasticity. In terms of company attributes, when the enterprise is a subsidiary company, the value of enterprise attributes is 1; otherwise, it is 0. In terms of industry characteristics, this paper refers to the industry classification methods proposed by Zeng and Wu [35] and judges whether they are consumer goods industries according to S&P Global Industry Classification Standard ("Yes"= 1; "No" = 0).

**Table 1.** Design and description of variables. CEO: chief executive officer.

| Type of Variable | Environment to Which It Belongs | Dimension of Variables | Name of Variables | Description of Variables | Questionnaire Number | Variable Assignment |
|---|---|---|---|---|---|---|
| Explained Variables | — — — — | Gender | CEO's gender | Is the CEO a woman? | B7a | "Yes" = 1; "No" = 0 |
| Explaining Variables | Micro environment | Basic Features | Enterprise size | Number of full-time employees | L1 | Logs of selected variables |
| | | | Enterprise age | When did the company start operating? | B5 | 2011—Logs of selected variables by adding 1 after the year of establishment |
| | | | Corporate attributes | Is the enterprise a branch of a certain company? | A7 | "Yes" = 1; "No" = 0 |
| | | | Industrial features | Judging whether it belongs to the consumer goods industry according to S&P Global Industry Classification Standard | a4a | "Consumer goods industry" = 1; "No" = 0 |
| | | Stock Equity Characteristics | Female owner | Does the company have a female owner? | B4 | "Yes" = 1; "No" = 0 |
| | | | Foreign shareholding | Proportion of foreign shareholding | B2b | Shareholding ratio |
| | | | State-owned shareholding | Proportion of state-owned shareholding | B2C | Shareholding ratio |
| | | Operating Features | Product market | Sales scope of products | E1 | Local = 1; Entire country = 2; International = 3 |
| | | | Financial risk | Sources of corporate funds | K3 | Proportion of self-possessed fund and retained earnings in all funds. The higher the proportion, the lower the financial risk. |

**Table 1.** *Cont.*

| Type of Variable | Environment to Which It Belongs | Dimension of Variables | Name of Variables | Description of Variables | Questionnaire Number | Variable Assignment |
|---|---|---|---|---|---|---|
| | Meso environment | Industrial environment | Competition order | Is the competitor a non-registered company? | E11 | "Legal enterprise" = 1; "Unregistered" = 0 |
| | | | Market environment | Only objective indicators are used to measure the depth and breadth of regional market-oriented reform | —— | The data were derived from the *Report on China's Provincial Marketization Index* (2016). Regarding the relevant literature, the score is in the mean value and above = 1, below the mean value = 0. |
| | Macro environment | Economic environment | Economic level | Measuring the level of economic development in such a region | —— | The data comes from the statistical yearbooks of different regions, and the variables are set as the logarithm of per-capita GDP. |
| | | | Radiation effect | Distances from central cities such as Beijing, Shanghai, and Guangzhou | —— | Using Google Maps to measure the nearest road distances to four central cities and conduct the normalization processing |
| | | Political environment | City level | The political level of the city in which it is located | A3 | Prefecture-level city = 1; Provincial capital city but not sub-provincial city = 2; Sub-provincial city = 3; Direct-controlled municipality = 4 |
| | | Cultural environment | Gender culture | Do you agree with the statement that female employees should be fired first in economic depression? | A42 | The data comes from CGSS 2012, reflecting the region's overall view of women. "Totally disagree" = 1; "Somewhat disagree" = 2; "It doesn't matter if I agree or disagree." = 3; "Partially Agree" = 4; and "Totally agree" = 5 |

The chief executive officer (CEO) is the supreme administrative officer responsible for the daily operation of an enterprise. This position is distinct from other senior management positions in ranking, power, influence, and so on. When the CEO's gender is female, the variable is defined as 1; otherwise, it is 0.

**Table 2.** Table of descriptive statistical analysis.

| Variables | Sampling Size | Average Value | Standard Deviation | Minimum Value | Maximum Value |
|---|---|---|---|---|---|
| CEO's gender | 2559 | 0.108 | 0.311 | 0 | 1 |
| Enterprise size | 2562 | 4.151 | 1.355 | 4 | 30,000 |
| Enterprise age | 2493 | 2.553 | 0.978 | 1 | 124 |
| Corporate attributes | 2562 | 0.132 | 0.339 | 0 | 1 |
| Industry characteristics | 2562 | 0.338 | 0.473 | 0 | 1 |
| Female ownership | 2562 | 0.608 | 0.488 | 0 | 1 |
| Foreign shareholding | 2555 | 0.039 | 0.171 | 0 | 1 |
| State-owned shareholding | 2555 | 0.032 | 0.158 | 0 | 0.95 |
| Product market | 1621 | 1.907 | 0.516 | 1 | 3 |
| Financial risk | 2510 | 0.889 | 0.206 | 0 | 1 |
| Competition order | 2467 | 0.448 | 0.497 | 0 | 1 |
| Market environment | 2562 | 0.497 | 0.500 | 0 | 1 |
| Economic level | 2562 | 11.147 | 0.300 | 10.595 | 11.612 |
| Radiation function | 2562 | 0.260 | 0.236 | 0 | 1 |
| City level | 2562 | 2.161 | 1.001 | 1 | 4 |
| Gender culture | 2562 | 2.131 | 0.128 | 1.788 | 2.420 |

In terms of stock equity characteristics, as an effective incentive means, equity promotes enterprise development [36]. Therefore, equity characteristics are one of the factors affecting CEO gender. This paper chooses three variables: female owners, foreign shareholding, and state-owned shareholding. Regarding female owners, the main measurement criterion is to see whether there are female owners ("Yes" = 1; "No" = 0). Foreign and state-owned shareholdings mainly reflect the company's shareholding situation, which is set as the proportion of foreign and state-owned shareholding.

In terms of operating characteristics, this paper sets two variables: product market and financial risk. The product market reflects the scope of the product sales of an enterprise, with local sales set as 1, and national and international sales set as 2 and 3, respectively. Overall, the women's risk preference is lower compared with the men's risk preference. Thus, this paper sets up financial risk variables to measure the impact of operating risk on the CEO's gender. Financial risk is mainly measured by the company's sources of funds, and it is set as the ratio of self-possessed funds and retained earnings in all funds. The higher the ratio, the lower the financial risk.

### 3.4.2. Meso-Environment: Industrial Environment

In terms of meso environment, this paper mainly considers the industrial environment as the main variable. The industrial environment includes competition order and market environment.

In terms of competition order, which has a profound impact on corporate operating performance and corporate governance [37,38], competition is the most prominent feature of the market economy, and it is an external feature that cannot be ignored. This paper refers to the research outcomes of Han, Cui, Chen, and Fu [9], and selects the legitimacy of competitors as the object of investigation. When the main competitor facing an enterprise is a legally registered enterprise, the variable assignment is set as 1. When it is an unregistered enterprise, the variable assignment is set as 0.

In terms of market environment, as China is in a period of transformation and development, the market environment will have a great impact on enterprise behaviors. Therefore, this paper refers to the practices proposed by Liu, Yang, and Yang [39], and sets the market environment variable in which only objective indicators are adopted to measure the depth and breadth of market-oriented reforms in various regions. The value assignment is 1 if scores are the average value and above. Otherwise, the assignment value is 0.

3.4.3. Macro-Environment: Regional Economy and Political and Cultural Environment

In this paper, the macro-environment is measured mainly from the aspects of the regional economy and the political and cultural environment.

In terms of economic environment, this paper mainly considers two aspects: economic level and radiation effect. The economic level is set as the logarithm of GDP per capita in the region. In terms of radiation function, central cities have a radiation effect on surrounding cities. The Chinese government made Beijing, Shanghai, Guangzhou, and Chongqing central cities to enhance their economic influence on neighboring cities. To investigate the economic radiation effects of these national central cities on other cities, the central distance variable is set in this paper. Since China's high-speed rail was still in the construction stage during 2011–2012, and the public acceptance level remains low, the central distance variable is thus set as the road distance between the company's location and Beijing, Shanghai, and Guangzhou, and normalized processing is conducted.

In terms of political environment, unlike Western countries, the Chinese government has classified major cities at different levels, meaning that the higher the level, the more attention and support from the central government. The city level affects a city's procurement of resources. Thus, the city level variable is set in this paper to measure the city's political status, and the higher level corresponds to a higher score.

In terms of culture, the influence of culture on people is imperceptible and lasting, and the process of CEO recruitment is influenced by gender culture. Therefore, the gender culture variable is set in this paper. The higher the score, the lower the gender equality, and the less favorable it is to females and vice versa.

## 4. Distribution Characteristics of the CEO's Gender in China

### 4.1. Geographical Distribution Characteristics of the CEO's Gender in China

In Figure 5, the distribution characteristics of female CEOs in China are reported. The black circle is used to represent the proportion of local female CEOs to all local CEOs. The larger the black circle, the higher the proportion of female CEOs in such regions will become. If observed from the north–south direction (Qinling-Huaihe is always considered as the demarcation between the north and the south of China. In this paper, samples of the south include: Shanghai, Nanjing, Wuxi, Suzhou, Nantong, Hefei, Hangzhou, Ningbo, Wenzhou, Guangzhou, Shenzhen, Foshan, Dongguan, Wuhan, and Chengdu; samples of the north include: Beijing, Shijiazhuang, Tangshan, Jinan, Qingdao, Yantai, Zhengzhou, Luoyang, Shenyang, Dalian, the same below), the distribution of female CEOs presents characteristics more in the south compared with the north, and there are more female CEOs in the Yangtze River Delta than in the Pearl River Delta. If observed from the east–west direction (in the east–west direction, China can be divided into three parts: east, middle, and west. The eastern part in this paper mainly refers to the eastern coastal areas of China, the central part mainly refers to the areas adjacent to the hinterland of the Central Plains, and the western area mainly refers to the provinces where western development policy is implemented. Samples from the eastern region include Shanghai, Nanjing, Wuxi, Suzhou, Nantong, Hefei, Hangzhou, Ningbo, Wenzhou, Guangzhou, Shenzhen, Foshan, Dongguan, Beijing, Jinan, Qingdao, Yantai, Shenyang, and Dalian. Samples from the central region include Wuhan, Shijiazhuang, Tangshan, Zhengzhou, and Luoyang. The western region mainly includes Chengdu, and the same is indicated below), the eastern coastal areas obviously have more female CEOs than in the central and western areas, showing the distribution characteristics to be more prevalent in the east than in the west. In summary, the distribution characteristics of the CEO's gender in China are more concentrated in the south and the east than in the north and the west.

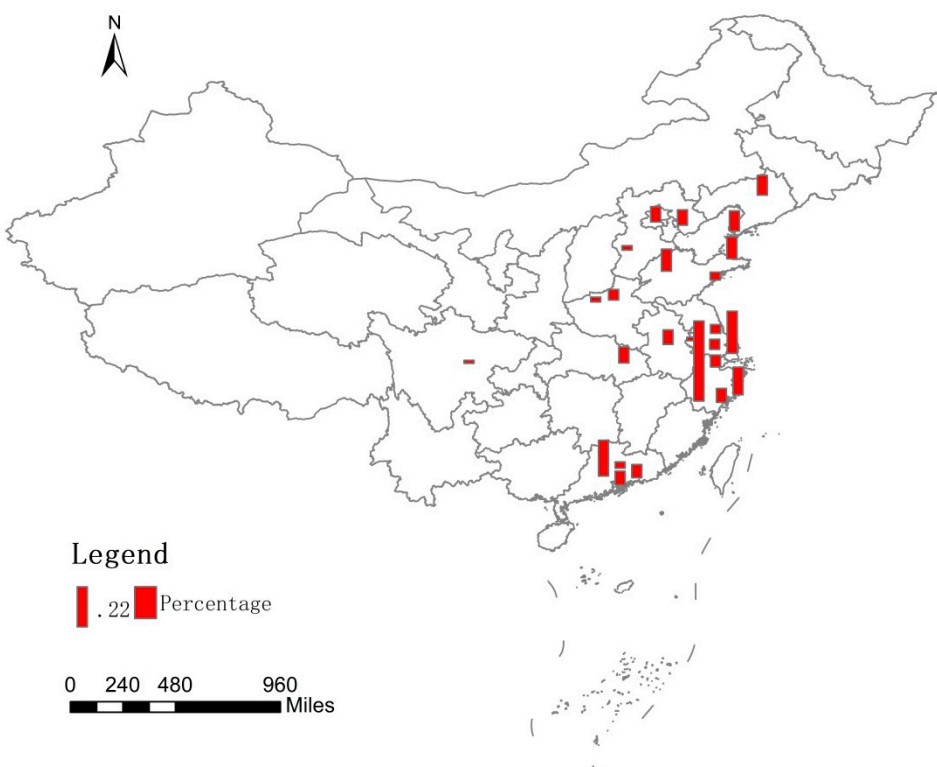

**Figure 5.** Distribution characteristics of the CEO's gender in China.

## 4.2. Micro-Environment and the CEO's Gender

To visualize the relationship between company characteristics and the CEO's gender, this paper classifies and summarizes CEO gender situations according to company micro-environment. The results are reported in Table 3. As the enterprise size, enterprise age, financial risk, and operating performance are continuous variables, this paper categorizes them into five groups. Regarding foreign and state-owned shareholding, this paper subdivides samples that gave the answer of "having" foreign or state-owned shareholding into five groups. Other variables are grouped according to the rules for variable assignment.

First, we assess the following basic company characteristics. (1) Enterprise size: the number and proportion of female CEOs have shown a downward trend with the constant expansion of enterprise size as indicated in the table above. Thus, in an enterprise with four to 17 employees, female CEOs are more likely to be promoted. (2) Enterprise age: the number and proportion of female CEOs in an enterprise with one to six years in top management. (3) Corporate attributes: subsidiaries have obvious advantages over non-subsidiaries. (4) Industry characteristics: the number and proportion of female CEOs in consumer goods industries are higher than those in non-consumer goods industries.

Second, we assess the corporate shareholding characteristics. (1) In enterprises with female owners, the number and proportion of female CEOs are significantly higher than in those without female owners. (2) Foreign shareholding: First, this paper compares the difference between having shareholding and not, and finds that the proportion of female CEOs in enterprises with foreign shareholding is significantly higher than those enterprises without. Second, this paper further compares the impacts of different shareholding proportions on CEO's gender, and finds that the proportion of female CEOs is the highest among companies with a shareholding ratio between 80–90%; (3) State-owned shareholding: First, the number and proportion of female CEOs in enterprises without state-owned shares are significantly higher than those enterprises having state-owned holdings. Second, in enterprises with state-owned shareholding, the lower the proportion is, the higher the number of female CEOs.

**Table 3.** Micro-environment and CEO's gender.

| Variables | Meaning of Variables | Male | Female | Summary | Proportion of Female CEOs |
|---|---|---|---|---|---|
| Enterprise size | 4–17 persons | 415 | 70 | 485 | 14.433% |
| | 18–48 persons | 470 | 63 | 533 | 11.820% |
| | 49–88 persons | 453 | 59 | 512 | 11.523% |
| | 89–180 persons | 474 | 42 | 516 | 8.140% |
| | 182–30,000 persons | 470 | 43 | 513 | 8.382% |
| | Summary | 2282 | 277 | 2559 | 10.825% |
| Enterprise age | 1–6 | 400 | 63 | 463 | 13.607% |
| | 7–8 | 352 | 40 | 392 | 10.204% |
| | 9–10 | 416 | 59 | 475 | 12.421% |
| | 11–14 | 513 | 57 | 570 | 10.000% |
| | 12–124 | 537 | 53 | 590 | 8.983% |
| | Summary | 2218 | 272 | 2490 | 10.924% |
| Corporate attributes | Non-subsidiary companies | 1992 | 228 | 2220 | 10.270% |
| | Subsidiaries | 290 | 49 | 339 | 14.454% |
| | Summary | 2282 | 277 | 2559 | 10.825% |
| Industry characteristics | Non-consumer goods industry | 1554 | 139 | 1693 | 8.210% |
| | Consumer goods industry | 728 | 138 | 866 | 15.935% |
| | Summary | 2282 | 277 | 2559 | 10.825% |
| Female ownership | Not have | 986 | 18 | 1004 | 1.793% |
| | Have | 1296 | 259 | 1555 | 16.656% |
| | Summary | 2282 | 277 | 2559 | 10.825% |
| Foreign shareholding | Not have | 2140 | 256 | 2396 | 10.684% |
| | Have | 137 | 19 | 156 | 12.179% |
| | 5–25% | 19 | 1 | 20 | 5.000% |
| | 30–40% | 34 | 7 | 41 | 17.073% |
| | 45–71 | 30 | 2 | 32 | 6.250% |
| | 80–90 | 3 | 1 | 4 | 25.000% |
| | 100% | 51 | 8 | 58 | 13.793% |
| | Summary | 2277 | 275 | 2552 | 10.776% |
| State-owned shareholding | Not have | 2172 | 272 | 2444 | 11.129% |
| | Have | 105 | 3 | 108 | 2.778% |
| | 5–50% | 14 | 1 | 15 | 6.667% |
| | 51–80% | 25 | 1 | 26 | 3.846% |
| | 85–88% | 21 | 0 | 21 | 0.000% |
| | 89–89% | 10 | 0 | 10 | 0.000% |
| | 90–95% | 35 | 1 | 36 | 2.778% |
| | Summary | 2282 | 277 | 2559 | 10.825% |
| Product market | Local | 274 | 24 | 298 | 8.054% |
| | Domestic | 1088 | 85 | 1173 | 7.246% |
| | International | 124 | 23 | 147 | 15.646% |
| | Summary | 1486 | 132 | 1617 | 8.163% |
| Financial risk | 100% | 1554 | 161 | 1715 | 9.388% |
| | 98–90% | 97 | 12 | 109 | 11.009% |
| | 89–77% | 204 | 23 | 227 | 10.132% |
| | 75–65% | 129 | 23 | 152 | 15.132% |
| | 60–35% | 170 | 37 | 207 | 17.874% |
| | 34–0% | 84 | 13 | 97 | 13.402% |
| | Summary | 2238 | 269 | 2507 | 10.730% |

Third, we assess the enterprise operating characteristics. (1) Product market: there are more female CEOs in enterprises whose scope of sales is located in their home country and significantly higher

than enterprises with a scope of sales located in their home country and abroad. As to proportion, the proportion of female CEOs in international enterprises with their scope of sales located abroad is significantly higher than those enterprises with a scope of sales located at home and in local areas. (2) Financial risk: A lower value means a higher financial risk. Thus, with the increase of financial risk, the proportion and number of female CEOs increase at first and then decrease, and reach the maximum when the financial risk falls in the range of 35% to 60%.

### 4.3. Meso-Environment and the CEO's Gender

The meso-environment and the CEO's gender distribution are reported in Table 4. Since competition order and market environment are both 0, 1 variables, this paper mainly groups according to the meaning of variables.

**Table 4.** Meso-environment and CEO's gender.

| Variables | Meaning of Variables | Male | Female | Summary | Proportion of Female CEOs |
|---|---|---|---|---|---|
| Competition order | Competitors' unregistered enterprises | 1214 | 147 | 1361 | 10.801% |
| | Competitors' unregistered enterprises | 986 | 117 | 1103 | 10.607% |
| | Summary | 2200 | 264 | 2464 | 10.714% |
| Market environment | Lower than the average value | 1171 | 118 | 1289 | 9.154% |
| | Higher than the average value | 1111 | 159 | 1270 | 12.520% |
| | Summary | 2282 | 277 | 2559 | 10.825% |
| | Summary | 2282 | 277 | 2559 | 10.825% |

First, in terms of competition order, the number of female CEOs in companies whose competitors are unregistered enterprises is higher than companies whose competitors are legally registered enterprises, and there is little difference in the proportion of female CEOs. Second, in terms of market environment, the number and proportion of female CEOs in those samples with a better-than-average market environment is significantly higher than those in the lower-than-average market environment. Thus, a good market environment is more conducive to the growth of female CEOs.

### 4.4. Macro-Environment and the CEO's Gender

Corporate macro characteristics and the CEO's gender distribution are reported in Table 5. Since the economic level, radiation effect, and gender culture are continuous variables, the above variables are divided into five groups in this paper.

First, we look at the economic environment. (1) Along with the improvement of economic level, the number and proportion of female CEOs showed a trend of increasing at first and then slightly decreasing, reaching the maximum when the per capita GDP was RMB 81,658–91,202. (2) In terms of radiation effect, as the distance to central cities increases, the proportion of female CEOs showed the trend of increasing at first and then decreasing, reaching the maximum 14.286% when the distance is between 135.6–174.5 km. Such distance is equivalent to that from Shanghai to Wuxi, Nantong, or Hangzhou, or from Guangzhou to Shenzhen. The minimum 8.606% is reached when the distance is in the range of 694 to 840.1 km. Such distance is equivalent to that from Beijing to Zhengzhou or Shenyang, or from Guangzhou to Wuhan.

**Table 5.** Macro environment and CEO's gender.

| Variables | Meaning of Variables | Male | Female | Summary | Proportion of Female CEOs |
|---|---|---|---|---|---|
| Economic level | RMB 39,919–48,755 | 515 | 31 | 546 | 5.678% |
| | RMB 56,005–68,315 | 471 | 48 | 519 | 9.249% |
| | RMB 70,380–76,263 | 491 | 54 | 545 | 9.908% |
| | RMB 81,658–91,202 | 312 | 78 | 390 | 20.000% |
| | RMB 91,295–110,421 | 493 | 66 | 559 | 11.807% |
| | Summary | 2282 | 277 | 2559 | 10.825% |
| Radiation effect | 0–102.1 km | 495 | 71 | 566 | 12.544% |
| | 135.6–174.5 km | 348 | 58 | 406 | 14.286% |
| | 176.7–304.5 km | 362 | 41 | 403 | 10.174% |
| | 410.8–661.9 km | 397 | 43 | 440 | 9.773% |
| | 694–840.1 km | 680 | 64 | 744 | 8.602% |
| | Summary | 2282 | 277 | 2559 | 10.825% |
| City level | Prefecture-level city | 905 | 72 | 977 | 7.369% |
| | Provincial capital city | 300 | 27 | 327 | 8.257% |
| | Sub-provincial city | 968 | 155 | 1123 | 13.802% |
| | Municipalities directly under the central government | 109 | 23 | 132 | 17.424% |
| | Summary | 2282 | 277 | 2559 | 10.825% |
| Gender culture | 1.787–1.988 | 307 | 80 | 387 | 20.672% |
| | 2.0025–2.0375 | 576 | 76 | 652 | 11.656% |
| | 2.11–2.194514 | 498 | 36 | 534 | 6.742% |
| | 2.202783–2.42 | 901 | 85 | 986 | 8.621% |
| | Summary | 2282 | 277 | 2559 | 10.825% |

Second, in terms of political environment, along with improvement at the city level, the number of female CEOs showed a trend of increasing at first and then slightly decreasing. However, considering the difference in sample sizes, the proportion of female CEOs presents the upward trend, and the proportion of female CEOs in municipalities directly under the central government is significantly higher than in other cities. Therefore, a higher city level is better for female CEOs.

Third, regarding gender culture, the higher the score, the more gender inequality and the more it is unfair to women. Therefore, in regions with the lowest gender culture score, the proportion of female CEOs will be higher. That is, the gender culture of such regions is fairer to women and more favorable for women to grow into CEOs.

To sum up, the general characteristics of enterprises with high CEOs gender scores are summarized as below in this paper.

1.  In terms of micro-environment, the enterprise has a small size, with four to 17 employees. It is relatively new and has been established within one to six years. It is a subsidiary company. It belongs to the consumer goods industry. It has female owners. It has 80–90% of foreign shareholding and no state-owned shareholding. The product sales are mainly focused on the international market. Its financial risk is high, and the proportion of its self-owned funds is within 35–60%.
2.  In terms of meso-environment, the company's competitors are mainly unregistered enterprises, and the company is in a better market environment than the average.
3.  In terms of macro-environment, the company is located in areas with good economic development level, with GDP per capita ranging between RMB 81,658–91,202. It is 135.6 to 174.5 km away from Beijing, Shanghai, Guangzhou, Chongqing, and other national central cities, and is located in municipalities directly under the central government. Moreover, it is located in areas with equal gender culture where women are treated more equally.

## 5. Empirical Analysis

To further test the impact of the corporate-level micro-environment, the industry-level meso environment, and the regional-level macro environment on the CEO's gender, this paper conducts empirical analysis. First, the impacts of the micro, meso, and macro environment on the CEO's gender are studied through the benchmark regression, and these three environments are analyzed within the same framework. Second, the moderating role of the meso-environment on the macro-environment is further analyzed, revealing the complexity of impacts of the macro-environment and meso-environment on the CEO's gender. Subsequently, through the application of marginal effect analysis, the extent to which the changes in environment factors may influence the CEO's gender is further studied. Finally, a robustness test is conducted in this paper.

### 5.1. Benchmark Regression

In this paper, Stata 14.0 is used for empirical analysis. Since the explained variable is binary variables of 0 and 1, logit model is thus used. Meanwhile, to reduce the influence of heteroscedasticity, robust regression analysis is used in the regression analysis of this paper. The outcomes of benchmark regression are reported in Table 6, while the impact outcomes of the micro-environment, meso-environment, and macro-environment on the CEO's gender are reported in models 1–3, and the overall impact outcomes of the micro-environment, meso-environment, and macro-environment on the CEO's gender are reported in Model 4. All the models in Table 6 are overall significant. Compared with models 1 to 3, the Chi-2 and pseudo R2 values of Model 4 are significantly improved, indicating that the degree of model optimization is continuously improved and the degree of model interpretation is continuously improved. The mechanism of how the environment affects the gender of CEOs is shown in Figure 6.

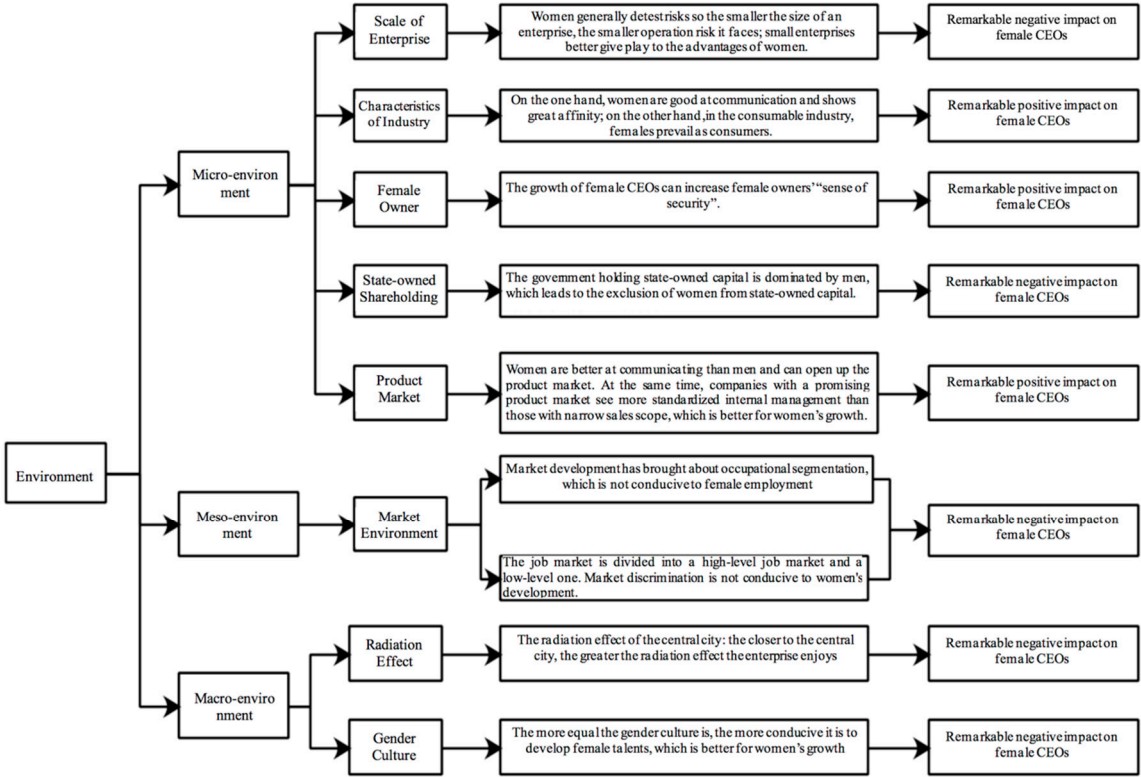

**Figure 6.** Schematic diagram of the mechanism of the environment's impact on CEO gender.

**Table 6.** Main empirical results of the impact of environment on CEO gender.

| Variables | Model 1 | Model 2 | Model 3 | Model 4 |
|---|---|---|---|---|
| Enterprise size | −0.203 ** | | | −0.219 ** |
| | (−2.52) | | | (−2.52) |
| Enterprise age | −0.102 | | | −0.0524 |
| | (−0.58) | | | (−0.29) |
| Corporate attributes | 0.264 | | | 0.207 |
| | (0.82) | | | (0.57) |
| Industry characteristics | 0.622 *** | | | 0.627 *** |
| | (3.13) | | | (2.95) |
| Female ownership | 2.257 *** | | | 2.393 *** |
| | (6.71) | | | (6.35) |
| Foreign shareholding | 0.638 | | | 0.426 |
| | (1.45) | | | (0.82) |
| State-owned shareholding | −1.658 * | | | −1.813 * |
| | (−1.90) | | | (−1.96) |
| Product market | 0.476 ** | | | 0.462 ** |
| | (2.46) | | | (2.41) |
| Financial risk | −0.689 * | | | −0.684 |
| | (−1.78) | | | (−1.52) |
| Competition order | | 0.0236 | | 0.0280 |
| | | (0.18) | | (0.13) |
| Market environment | | 0.319 ** | | −0.764 * |
| | | (2.43) | | (−1.81) |
| Economic level | | | 0.205 | 0.155 |
| | | | (0.79) | (0.38) |
| Radiation effect | | | −1.013 *** | −1.973 *** |
| | | | (−3.12) | (−2.60) |
| City level | | | 0.296 *** | 0.142 |
| | | | (4.33) | (1.19) |
| Gender culture | | | −1.017 * | −2.729 ** |
| | | | (−1.69) | (−2.53) |
| Constant term | −3.622 *** | −2.296 *** | −2.681 | 0.822 |
| | (−5.01) | (−19.78) | (−0.79) | (0.15) |
| Chi$^2$ | 83.55 *** | 5.90 * | 52.43 *** | 103.75 *** |
| Pseudo R$^2$ | 0.128 | 0.0035 | 0.0299 | 0.153 |
| Log-pseudolikelihood | −373.675 | −836.056 | −851.085 | −339.321 |
| Sample size | 1534 | 2464 | 2559 | 1465 |

Note: The values in parentheses are *t* values; *, **, and *** sub-tables represent significance at 10%, 5%, and 1% levels.

### 5.1.1. Micro-Environment

Basic Features

There is a significantly negative correlation between the enterprise size and the CEO's gender, that is, the smaller an enterprise remains, the greater possibility that female CEOs may be hired. Perhaps the smaller the enterprise size, the less operational risk it may face, which is just in line with women's low risk appetite.

There is no significantly positive correlation between the enterprise age and the CEO's gender. Considering that the coefficient is too small, and the enterprise age has little impact on the CEO's gender, the reason why the impact coefficient is positive may be that the longer the company operates, the more standardized its internal management system may become, which provides a better environment for female CEOs.

In terms of corporate attributes, subsidiaries have no significant impact on the CEO's gender, yet its influencing coefficient is positive.

Regarding industry characteristics, the consumer goods industry significantly impacts the CEO's gender, i.e., compared with other industries, companies in the consumer goods industry are more likely to choose female CEOs. This is mainly because of industry variance. The consumer goods industries are dominated by clothing, leisure, media, daily products, food, drug retail, and so on. Thus, the consumer goods industries are more family-oriented, and female CEOs are more aware of the needs of family and better at communicating with customers than male CEOs.

Stock Equity Characteristics

Female ownership significantly promoted the employment of female CEOs, raising the CEO's gender. This is mainly because of female owners' mentality of "penguin huddle" toward other females. At the top management of a company, females are a scarcity, and the increase in the same gender may enhance the sense of "security" among female owners [2]. The actual situation in China also affirmed this. According to the survey data of Chinese women entrepreneurs, in female-led enterprises, the proportion of female employees and female mid-level and senior-level managers exceeds that of male-led enterprises [40], and the extant literature indicates that female directors have significantly positive impacts on the increase of female senior managers [33].

Foreign shareholding has no significant impact on the CEO's gender, while state-owned shareholding has a significantly negative impact on the CEO's gender; that is, the lower the proportion of state-owned shareholding, the more likely it is for the company to choose male CEOs. This is mainly because government officials holding state-owned capitals are generally male officials, which leads to the exclusion of females from state-owned capitals.

Operating Characteristics

In terms of the product market, it has a significantly positive impact on the CEO's gender. That is, the broader the product sales scope, the more likely it is for the company to choose female CEOs. This is mainly because female CEOs are better at communication compared with male CEOs and can open up the product market more efficiently.

In terms of financial risk, there is a significantly negative correlation between the company's financial risk and the CEO's gender. Considering the value assignment of financial risk variables, this indicates that the higher the company's financial risk, the more likely it is to choose a female CEO. Women have lower risk preferences, so high-risk companies may choose low-risk females to defuse risks.

5.1.2. Meso-Environment

Competition Order

The competition order has a negative impact on the CEO's gender, but it is not significant. This is mainly related to the women's personality. Studies have found that female senior managers may significantly reduce corporate violations [41], so companies in industries rife with illegal competition are unlikely to choose female CEOs.

Market Environment

The impact of market environment variables on the CEO's gender is more complicated. In Model 2, it was found that the market environment has a significantly positive impact on the CEO's gender; that is, the better the market environment, the more favorable it is for females to become CEOs. In Model 4, the market environment has a significantly negative impact on the CEO's gender; that is, the better the market environment, the more unfavorable for women to become CEOs. Generally, the better the market environment, the fairer the market will be, and the friendlier to women it will become.

Then, why does the market environment have a remarkably negative impact on female CEOs? This essay holds that there are two different mechanisms, as shown in Figure 6. First of all, the division

of labor in the market brings about occupational segregation, which is not conducive to women's growth. As Adam Smith emphasized in *The Wealth of Nations*, the division of labor drives the economy. Therefore, with the continuous development of the economy, the division of labor in the market is continuously becoming specific and specialized. However, due to the physiological differences between women and men, some occupations are more suitable for men, not women, such as steel mill workers, maritime seafarers, etc., while secretarial, teachers, and other occupations are more suitable for women. In the long run, the market has weakened the professional status of women, while drawing on the advantages of men, which is solidified through the market system. This causes the occupational segregation of women as a result.

Second, the differentiation of the labor market shapes discrimination against women. The labor market theory holds that the labor market is not a simple unified market, but rather a differentiated market. The labor market can be divided into a high-level labor market and a secondary labor market. The high-level labor market is composed of people with a high level of education and technical expertise, and the secondary labor market is composed of people with no technical expertise. The differentiation of the labor market refers to the mutual exclusion of different labor markets. It is impossible to enter the high-level labor market from the secondary labor market, and the high-level labor market excludes the secondary labor market. Due to gender discrimination, women are less educated in the early stages of market development and do not have the ability to enter the advanced labor market. From then on, it is relatively difficult for women to enter the advanced labor market. In the secondary labor market, according to institutional economics theory, the secondary labor market is a market of incomplete information, and market entities decide recruitment according to gender, race, education level, and professional experience. When people generally believe that women's education levels are lower than men's and women do not need professional development, women face more discrimination. Therefore, gender is one of the signs of labor market segregation, and it influences different classified labor markets, thus putting women in a weak position.

As for the reason why two different results appeared, we believe that in this paper, there are other variables that regulate the impact of market environment on the CEO's gender, and this kind of moderating role will be further discussed in a later section.

### 5.1.3. Macro-Environment

#### Economic Environment

In terms of economic level, there is no significant correlation between economic level and the CEO's gender, but the economic level has a positive impact on the CEO's gender: that is, the higher the economic level, the more likely it is for the company to choose a female CEO. The economic level can reflect the overall situation of regional development. In areas with higher economic levels, systems are relatively sound, and thus conducive to the promotion of the CEO's gender.

In terms of radiation effect, the radiation effect has a significantly negative impact on the CEO's gender. That is, the closer the company is located to central cities such as Beijing, Shanghai, Guangzhou, and Chongqing, the more likely it is to choose female CEOs. This may be because the closer the company is to central cities, the more radiation it may receive, and the friendlier it is to women.

#### Political Environment

There is a significantly positive correlation between urban political level and the CEO's gender, which indicates that the higher the urban political level, the more likely it is for the company to choose a female CEO. The city level is a common phenomenon in China, and the Chinese government allocates resources by determining city levels. The higher the city level, the more resources it may acquire. Subsequently, the increase of city population is driven and systems are improved, which in turn helps women receive promotions and increases the CEO gender ratio.

Cultural Environment

In terms of cultural environment, there is a significantly negative correlation between gender culture and the CEO's gender, and such a negative direction is due to the variable assignment. That is, the more females are respected in gender culture, the more equal it is to females and the easier it is for women to become CEOs, as it is mainly because the equal cultural environment provides a broad space for women's development and reduces discrimination against women.

### 5.2. Further Analysis: The Moderating Role of Macro Environment in Meso Environment

In models 2 and 4 of Table 6, there are two opposite outcomes of market environment on the CEO's gender. In fact, the more perfect the market environment is, the more favorable it is for women to become CEOs. Therefore, it is argued in this paper that other variables might regulate the impact of market environment on the CEO's gender, which needs to be further studied.

As shown in Figure 7, the gender culture in the macro-environment adjusts the market environment in the meso-environment, mainly adjusting the influence of the market environment on the gender of CEO by respecting and understanding gender differences, recognizing women's social values, and promoting women's socialization.

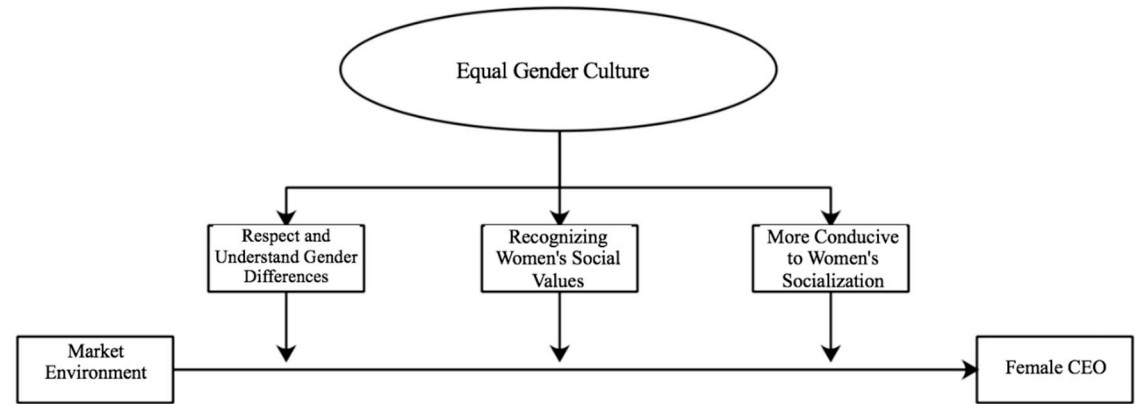

**Figure 7.** Illustration of how the gender culture adjusts the market environment.

First, an equal gender culture is conducive to creating an atmosphere of respecting and understanding gender differences. The differences in physiological functions between men and women and the social values behind them are the basis of market discrimination. In an equal gender culture, men's willingness to listen, recognize, understand, and accept women's physiological differences will largely help develop women's talents.

Second, an equal gender culture helps enterprises recognize women's social values. The reason enterprises are reluctant to hire women lies in the conflict between women's social values and corporate values. Women have the social function and value of breeding offspring, which is not available to men and only women can do it. Therefore, when a woman gives birth, her family and social value is increased. However, a female employee's giving birth does not directly increase the corporate value. On the contrary, the mandatory provisions such as maternity leave and maternity benefits have a negative impact on the company. Therefore, the company reduces female employment. However, when the gender culture is more and more equal, the company actively carry out a sense of social responsibility and assumes the necessary corporate social responsibility [42,43]. Recognizing the social value of women can benefit women's growth.

Finally, an equal gender culture is more conducive to the socialization of women and the development of female talents. Socialization is a process in which women learn from and integrate into the surrounding men. The external environment makes new requirements for women's development. Women are socialized through learning mechanisms, interactive mechanisms, and educational

mechanisms to finish the process of learning from men and gradually adapt to the requirements of the external environment. Therefore, the more equal the gender culture of an enterprise is, the more women can display their unique charm and develop their talents.

A previous study conducted by Han, Cui, Chen, and Fu found that gender culture significantly regulated the impact of female CEOs on corporate innovation behaviors [9]. Thus, by referring to the practices of such literature, this paper studies the gender culture of the cultural environment as a moderating variable. The empirical results are reported in Table 7, and the model is significant as a whole. Models 5 and 6 are empirical results after increasing the moderating role of gender culture. To facilitate comparison, we placed models 2 and 4 from Table 6 into Table 7.

**Table 7.** Empirical results of the moderating effect of gender culture on market environment.

| Variables | Model 2 | Model 5 | Model 4 | Model 6 |
|---|---|---|---|---|
| Competition order | 0.0236 | −0.0840 | 0.0280 | 0.0183 |
| | (0.18) | (−0.63) | (0.13) | (0.08) |
| Market environment | 0.319 ** | 12.06 *** | −0.764 * | 2.722 |
| | (2.43) | (4.77) | (−1.81) | (0.51) |
| Gender culture | | 0.251 | −2.729 ** | −2.119 |
| | | (0.33) | (−2.53) | (−1.50) |
| Market environment * Gender culture | | −5.692 *** | | −1.622 |
| | | (−4.74) | | (−0.66) |
| Enterprise size | | | −0.219 ** | −0.218 ** |
| | | | (−2.52) | (−2.52) |
| Enterprise age | | | −0.0524 | −0.0539 |
| | | | (−0.29) | (−0.30) |
| Company attributes | | | 0.207 | 0.203 |
| | | | (0.57) | (0.56) |
| Industry characteristics | | | 0.627 *** | 0.636 *** |
| | | | (2.95) | (2.98) |
| Female ownership | | | 2.393 *** | 2.378 *** |
| | | | (6.35) | (6.28) |
| Foreign shareholding | | | 0.426 | 0.416 |
| | | | (0.82) | (0.79) |
| State-owned shareholding | | | −1.813 * | −1.787 * |
| | | | (−1.96) | (−1.91) |
| Product market | | | 0.462 ** | 0.443 ** |
| | | | (2.41) | (2.33) |
| Financial risk | | | −0.684 | −0.639 |
| | | | (−1.52) | (−1.39) |
| Economic level | | | 0.155 | 0.325 |
| | | | (0.38) | (0.63) |
| Radiation effect | | | −1.973 *** | −1.608 * |
| | | | (−2.60) | (−1.79) |
| City level | | | 0.142 | 0.106 |
| | | | (1.19) | (0.81) |
| Constant term | −2.296 *** | −2.789 * | 0.822 | −2.438 |
| | (−19.78) | (−1.66) | (0.15) | (−0.32) |
| Chi$^2$ | 5.90 * | 42.99 *** | 103.75 *** | 103.69 *** |
| Pseudo R$^2$ | 0.0035 | 0.0242 | 0.1528 | 0.1534 |
| Log-pseudolikelihood | −836.056 | −818.670 | −339.321 | −339.097 |
| Sample size | 2464 | 2464 | 1465 | 1465 |

Note: The values in parentheses are *t* values; *, **, and *** sub-tables represent significance at 10%, 5%, and 1% levels.

In Model 5, the moderating variable of gender culture is added. Compared with Model 2, the coefficient of the market environment variable is significantly improved, and the interaction items are significantly negative (mainly due to the value assignment structure of the gender culture variable), showing that the more fair the environment that the gender culture enjoys, the more likely it is for a

company to choose female CEOs, increasing the CEO gender ratio. Thus, we find that the interaction between the meso-environment and macro environment can significantly promote the CEO's gender. In Model 6, the moderating variable of gender culture is also added. Compared with Model 4, the market environment variable changes from negative to positive, but the impact is not significant.

This illustrates two issues. First, the environment in which a company is placed does not affect the CEO's gender separately and directly. Instead, they affect the CEO's gender through the interaction between environments, such as moderating effect. Second, the gender culture in the macro-environment does not necessarily have a significant impact on the CEO's gender and the effective supervisions of government and other departments are required. For example, in models 4 and 6, the gender culture has an insignificant impact on the moderating effect of the market environment, but it changed the influence symbol of the market environment in Model 4. As a type of soft restraint and informal system, a culture does not impose mandatory restraint on corporate behavior in relation to market-chasing interests. This also explains why Germany has passed a bill mandating that the proportion of female CEOs in large enterprises should not be less than 30% (China News Network. Germany passed the law on the proportion of female CEOs in enterprises, stipulating that the proportion of female CEOs shall not be less than 30%. [EB/OL]. http://www.chinanews.com/gj/2015/03-06/7109101.shtml, 2015-03-06/2018-08-20). Similarly, the Chinese government has also issued the *Notice on Further Normalizing Recruitment Behavior to Promote Women's Employment*, further elaborating specific manifestations of gender discrimination in employment and requiring enterprises not to restrict women's employment and refuse to hire women on the grounds of gender (*Notice on Standardizing Recruitment Behavior and Promoting Women's Employment* issued by the Network of the Central People's Government of the People's Republic of China, the Ministry of Human Resources and Social Security and other nine departments [EB/OL]. http://www.gov.cn/xinwen/2019-02/25/content_5368180.htm).

### 5.3. Marginal Effect Analysis

To visually obtain the impact of the micro-environment, meso-environment, and macro-environment on the CEO's gender, the marginal effect analysis was conducted in this paper, and the results thereof were reported in Table 8. Among them, the impacts of gender culture on the marginal effect under the moderation of the market environment are reported in models 8 and 12.

### 5.3.1. Micro-Environment

In the micro-environment, the marginal effects of enterprise size, industry characteristics, female ownership, state-owned shareholding, and product market are significant.

In terms of basic company characteristics, the CEO's gender decreased by 0.0138% for every one unit added to the enterprise size. In terms of industry characteristics, the CEO's gender in the consumer goods industry increased by 0.0409% if compared with the non-consumer goods industry.

In terms of the stock equity characteristics, for every one unit added to the female ownership of a company, the CEO's gender increased by 0.156%. For every one unit decrease in state-owned shareholding, the CEO's gender increased by 0.118%. In terms of the operating characteristics of a company, for every one unit added to the product market of a company, the CEO's gender increased by 0.0301%.

**Table 8.** Analysis of marginal effects of environment on CEO gender.

| Variables | Model 7 | Model 8 | Model 9 | Model 10 | Model 11 | Model 12 |
|---|---|---|---|---|---|---|
| Enterprise size | −0.0138 ** (0.00552) | | | | −0.0143 ** (0.00570) | −0.0142 ** (0.00567) |
| Enterprise age | −0.00694 (0.0121) | | | | −0.00342 (0.0118) | −0.00351 (0.0118) |
| Company attributes | 0.0180 (0.0219) | | | | 0.0135 (0.0237) | 0.0132 (0.0237) |
| Industry characteristics | 0.0424 *** (0.0136) | | | | 0.0409 *** (0.0140) | 0.0414 *** (0.0141) |
| Female ownership | 0.154 *** (0.0246) | | | | 0.156 *** (0.0262) | 0.155 *** (0.0263) |
| Foreign shareholding | 0.0436 (0.0300) | | | | 0.0278 (0.0340) | 0.0271 (0.0342) |
| State-owned shareholding | −0.113 * (0.0598) | | | | −0.118 * (0.0608) | −0.116 * (0.0612) |
| Product market | 0.0325 ** (0.0132) | | | | 0.0301 ** (0.0125) | 0.0289 ** (0.0124) |
| Financial risk | −0.0470 * (0.0267) | | | | −0.0446 (0.0296) | −0.0416 (0.0301) |
| Competition order | | 0.00225 (0.0126) | −0.00788 (0.0125) | | 0.00183 (0.0141) | 0.00119 (0.0140) |
| Market environment | | 0.0304 ** (0.0126) | 1.132 *** (0.240) | | −0.0498 * (0.0278) | 0.177 (0.344) |
| Economic level | | | | 0.0193 (0.0246) | 0.0101 (0.0269) | 0.0212 (0.0335) |
| Radiation effect | | | | −0.0956 *** (0.0307) | −0.129 ** (0.0504) | −0.105 * (0.0595) |
| City level | | | | 0.0279 *** (0.00649) | 0.00927 (0.00781) | 0.00693 (0.00857) |
| Gender culture | | | 0.0235 (0.0719) | −0.0959 * (0.0566) | −0.178 ** (0.0716) | −0.138 (0.0931) |
| Market environment * Gender culture | | | −0.534 *** (0.114) | | | −0.106 (0.160) |
| Sample size | 1534 | 2464 | 2464 | 2559 | 1465 | 1465 |

Note: The values in parentheses are standard deviations; *, **, and *** sub-tables represent significance at 10%, 5%, and 1% levels.

### 5.3.2. Meso-Environment

In terms of meso-environment, the marginal effect of the market environment is significant, but the impacts vary in different models.

In Model 8, the market environment has a significant impact on the CEO's gender, and the company's probability of selecting a female CEO increased by 0.0304% for every one unit added to the market environment. In Model 9, under the moderating effect of gender culture, the CEO's gender increased by 1.132% for every one unit added to the market environment. In Model 11, the market environment has a negative impact on the CEO's gender, and the company's probability of selecting a female CEO decreased by 0.0498% for every one unit added to the market environment. In Model 12, under the moderating effect of gender culture, the impact of the market environment on the CEO's gender changed from negative to positive.

### 5.3.3. Macro Environment

In the macro environment, the marginal effects of radiation effect and gender culture are significant. In terms of radiation effect, the CEO's gender decreased by 0.129% for every one unit reduced in the

distance from company to central cities. In terms of gender culture, the CEO's gender increased by 0.178% for every one unit added to the gender culture.

*5.4. Handling of Endogenous Problems*

In terms of endogenous problems, this paper tries every possible method to minimize the impact of endogenous variables on research results. Firstly, the meso-environment and macro-environment belong to exogenous factors. Generally speaking, the behaviors of a company are unable to affect the development of the entire industry, and it is even less likely to directly affect the political level and economic and cultural level of this region. Thus, endogenous problems in the meso-environment and macro-environment are greatly weakened. However, in the micro-environment, there may be a two-way causal relationship between CEO genders and the micro-environment under which enterprises are located [44]. For example, along with corporate development, the construction of enterprise systems is becoming more and more perfect, which is in turn conducive to females becoming CEOs. There may be various other reasons such as the internal mutual supervision mechanism within the company [45], CEO power [46], enterprise size [22], etc. In addition, endogenous problems may also be caused by missing variables, measurement errors, and other reasons. Therefore, endogenous problems also need to be alleviated in this paper.

In this paper, the method of adding as many control variables as possible that affect both explanatory variables and dependent variables is adopted for the reasons as follows. (1) This method is an effective way to alleviate endogeneity, and it is adopted in international top-level journals. In this paper, the research results of Schijven and Hitt, [47] and Foss et al. [48] are used as reference, and the method of adding control variables is used to alleviate endogeneity. (2) In the aspect of variable design in this paper, in order to reflect the impact of the corporate micro-environment on CEO genders, it is not available or possible to choose a core variable, because no one variable can reflect the company's overall level. That is to say, a core variable can not reflect the overall picture of the micro-environment. Therefore, the selection of enterprise scale and corporate equity structure as core variables can not reflect the actual situation of the corporate micro-environment. In conclusion, this method is chosen to alleviate endogeneity in this paper.

The specific practice of using this method is to increase the control variables correlated to both dependent variables and independent variables. The endogeneity in this paper is caused by variables in the corporate micro-environment. For this reason, we have added more corporate micro-environments as the control variable. The variables and setting methods are specified as follows. (1) The degree of informationization (in questionnaire number CNo8, the question is: currently, what percentage of this establishment's workforce regularly use computers in their jobs?) considers there have been rapid developments in the degree of informationization in recent years. Thus, such a variable reflects the degree of corporate informationization, and this variable is set as the proportion of computers used in the work. The higher the proportion is, the higher the degree of corporate informationization will become. (2) Cooperative research and development (R&D) (in questionnaire number CNo5, the question is: in the last three years, did this establishment spend money on research and development activities contracted with other companies? The answer options are YES and No.) is set as a dummy variable that mainly reflects the cooperative R&D situation of enterprises. Cooperative R&D is assigned a 1; otherwise, 0. (3) Technology acquisition (in questionnaire number E6, the question is: does this establishment at present use technology licensed from a foreign-owned company, excluding office software?) is set as a virtual variable that is mainly reflected in the situation of corporate technology acquisition. The fact of technology acquisition is assigned a 1; otherwise, it is 0. (4) International certification (in questionnaire B8, the question is: does this establishment have an internationally recognized quality certification? (INTERVIEWER: If there is need for clarification, some examples are: ISO 9000 or 14,000, or HACCP)) is set as a virtual variable, and enterprises are assigned a 1 if they have passed international certification; otherwise, they are assigned a 0.

In Table 9, the empirical results of the endogeneity test are reported. From the results, all of the models are significant as a whole. In terms of specific variables, except that the market environment in Model 14 is not significant, other variables are significant, and there are no significant changes in the market environments in Model 16 and Model 17. Therefore, we can think that the endogenous problems have been effectively alleviated.

**Table 9.** Empirical analysis of endogenous tests.

|  | Model 13 | Model 14 | Model 15 | Model 16 | Model 17 |
|---|---|---|---|---|---|
| Enterprise size | −0.162 ** (−2.02) |  |  | −0.156 * (−1.76) | −0.157 * (−1.77) |
| Enterprise age | −0.0623 (−0.33) |  |  | −0.00792 (−0.04) | −0.00927 (−0.05) |
| Company attributes | 0.119 (0.34) |  |  | 0.106 (0.27) | 0.106 (0.27) |
| Industry characteristics | 0.600 *** (2.91) |  |  | 0.614 *** (2.78) | 0.619 *** (2.79) |
| Female ownership | 2.285 *** (6.35) |  |  | 2.322 *** (6.05) | 2.315 *** (6.00) |
| Foreign shareholding | 0.644 (1.40) |  |  | 0.568 (1.04) | 0.558 (1.01) |
| State-owned shareholding | −1.780 ** (−2.09) |  |  | −1.844 ** (−2.08) | −1.831 ** (−2.06) |
| Product market | 0.437 ** (2.18) |  |  | 0.416 ** (2.10) | 0.407 ** (2.07) |
| Financial risk | −0.639 (−1.52) |  |  | −0.637 (−1.31) | −0.615 (−1.25) |
| Competition order |  | −0.0332 (−0.16) |  | 0.0140 (0.06) | 0.00823 (0.04) |
| Market environment |  | 0.227 (1.14) |  | −0.862 ** (−1.96) | 0.997 (0.18) |
| Economic level |  |  | 0.295 (0.79) | 0.179 (0.42) | 0.262 (0.50) |
| Radiation effect |  |  | −0.935 * (−1.91) | −2.512 *** (−3.19) | −2.292 ** (−2.46) |
| City level |  |  | 0.154 (1.52) | 0.168 (1.35) | 0.149 (1.06) |
| Gender culture |  |  | −1.826 ** (−2.15) | −2.915 *** (−2.58) | −2.588 * (−1.84) |
| Market environment * Gender culture |  |  |  |  | −0.861 (−0.34) |
| Informationization | 1.086 ** (2.36) | 1.194 *** (2.82) | 0.845 ** (1.97) | 0.710 (1.44) | 0.719 (1.45) |
| Cooperative R&D | −0.0367 (−0.11) | −0.210 (−0.67) | −0.374 (−1.20) | −0.140 (−0.38) | −0.148 (−0.39) |
| Technology acquisition | 0.0641 (0.25) | 0.130 (0.56) | 0.0458 (0.21) | −0.255 (−0.93) | −0.253 (−0.92) |
| International certification | −0.302 (−1.30) | −0.379 * (−1.72) | −0.479 ** (−2.29) | −0.268 (−1.06) | −0.261 (−1.02) |
| Constant term | −3.995 *** (−5.34) | −2.679 *** (−10.17) | −1.842 (−0.38) | 0.855 (0.15) | −0.809 (−0.10) |
| Chi$^2$ | 97.72 | 12.50 * | 33.56 *** | 110.91 *** | 111.62 *** |
| Pseudo R$^2$ | 0.1359 | 0.0144 | 0.0359 | 0.1606 | 0.1608 |
| Log-pseudolikelihood | −356.71718 | −399.76577 | −425.34368 | −325.83632 | −325.77685 |
| Sample size | 1495 | 1497 | 1571 | 1435 | 1435 |

Note: The values in parentheses are *t* values; *, **, and *** sub-tables represent significance at 10%, 5%, and 1% levels.

### 5.5. Robustness Test

Based on previous empirical results, the following methods are used to test robustness.

#### 5.5.1. Consideration of Replacing Empirical Methods

In this paper, the regression model is replaced by the probit model. The empirical results are reported in Table 10, and the moderating effect of gender culture on market environment is reported in models 18 and 22. In terms of empirical results, the model as a whole is significant, and there is little difference with the benchmark regression results in Table 6. Thus, the empirical results are robust.

**Table 10.** Main empirical results after replacing empirical methods.

| Variables | Model 18 | Model 19 | Model 20 | Model 21 | Model 22 |
|---|---|---|---|---|---|
| Enterprise size | −0.109 ***<br>(−2.59) | | | −0.119 ***<br>(−2.66) | −0.119 ***<br>(−2.65) |
| Enterprise age | −0.0539<br>(−0.58) | | | −0.0229<br>(−0.24) | −0.0223<br>(−0.24) |
| Company attributes | 0.147<br>(0.89) | | | 0.127<br>(0.70) | 0.123<br>(0.68) |
| Industry characteristics | 0.327 ***<br>(3.16) | | | 0.341 ***<br>(3.10) | 0.344 ***<br>(3.12) |
| Female ownership | 1.061 ***<br>(7.67) | | | 1.127 ***<br>(7.35) | 1.121 ***<br>(7.28) |
| Foreign shareholding | 0.334<br>(1.46) | | | 0.234<br>(0.89) | 0.231<br>(0.88) |
| State-owned shareholding | −0.794 **<br>(−2.06) | | | −0.904 **<br>(−2.21) | −0.894 **<br>(−2.18) |
| Product market | 0.252 **<br>(2.53) | | | 0.243 **<br>(2.46) | 0.234 **<br>(2.38) |
| Financial risk | −0.420 **<br>(−2.00) | | | −0.387<br>(−1.64) | −0.368<br>(−1.54) |
| Competition order | | 0.0140<br>(0.20) | | 0.0169<br>(0.15) | 0.0115<br>(0.10) |
| Market environment | | 0.166 **<br>(2.43) | | −0.440 **<br>(−2.08) | 1.125<br>(0.42) |
| Gender culture | | | −0.504 *<br>(−1.66) | −1.601 ***<br>(−2.87) | −1.323 *<br>(−1.82) |
| Economic level | | | 0.111<br>(0.82) | 0.0746<br>(0.35) | 0.146<br>(0.57) |
| Radiation effect | | | −0.537 ***<br>(−3.18) | −1.117 ***<br>(−2.93) | −0.959 **<br>(−2.13) |
| City level | | | 0.157 ***<br>(4.29) | 0.0747<br>(1.21) | 0.0592<br>(0.89) |
| Constant term | −1.838 ***<br>(−5.10) | −1.333 ***<br>(−22.32) | −1.636<br>(−0.93) | 0.944<br>(0.34) | −0.479<br>(−0.12) |
| Market environment *<br>Gender culture | | | | | −0.728<br>(−0.59) |
| Chi$^2$ | 93.02 *** | 5.92 * | 50.23 *** | 114.69 *** | 114.38 *** |
| Pseudo R$^2$ | 0.1286 | 0.0035 | 0.0298 | 0.1565 | 0.1569 |
| Log-pseudolikelihood | −373.23996 | −836.05133 | −851.19876 | −337.8409 | −337.6714 |
| Sample size | 1534 | 2464 | 2559 | 1465 | 1465 |

Note: The values in parentheses are *t* values; *, **, and *** sub-tables represent significance at 10%, 5%, and 1% levels.

### 5.5.2. Considering the Impact of Biased Geographical Location of the Sample

Due to data reasons, the samples in this paper are mainly concentrated in the eastern coastal areas of China. Thus, considering the impact of uneven sample distribution, variables in the eastern region are set in this paper; that is, the value assignment of samples from the eastern region is 1 and the value assignment of samples from non-eastern region is 0. The empirical results are reported in Table 11. From the results, the samples in the eastern region are insignificant, so we can exclude the impact of sample concentration in the eastern region from the empirical results.

**Table 11.** Main empirical results after considering the geographical distribution of samples.

| Variables | Model 23 | Model 24 | Model 25 | Model 26 | Model 27 |
|---|---|---|---|---|---|
| Enterprise size | −0.110 *** (−2.61) | | | −0.114 ** (−2.54) | −0.114 ** (−2.54) |
| Enterprise age | −0.0537 (−0.58) | | | −0.0179 (−0.19) | −0.0180 (−0.19) |
| Company attributes | 0.153 (0.93) | | | 0.117 (0.65) | 0.117 (0.65) |
| Industry characteristics | 0.327 *** (3.15) | | | 0.338 *** (3.07) | 0.339 *** (3.06) |
| Female ownership | 1.058 *** (7.63) | | | 1.124 *** (7.32) | 1.122 *** (7.29) |
| Foreign shareholding | 0.344 (1.50) | | | 0.237 (0.90) | 0.236 (0.90) |
| State-owned shareholding | −0.780 ** (−2.01) | | | −0.922 ** (−2.25) | −0.918 ** (−2.23) |
| Product market | 0.247 ** (2.49) | | | 0.230 ** (2.35) | 0.228 ** (2.34) |
| Financial risk | −0.418 ** (−2.00) | | | −0.345 (−1.45) | −0.342 (−1.43) |
| Competition order | | 0.0107 (0.16) | | 0.00390 (0.03) | 0.00313 (0.03) |
| Market environment | | 0.0946 (1.15) | | −0.539 ** (−2.29) | −0.0798 (−0.03) |
| Economic level | | | 0.139 (1.02) | 0.130 (0.59) | 0.147 (0.57) |
| Radiation effect | | | −0.204 (−0.76) | −0.858 ** (−1.99) | −0.830 * (−1.81) |
| City level | | | 0.153 *** (4.20) | 0.0605 (0.96) | 0.0570 (0.86) |
| Gender culture | | | −0.672 ** (−2.04) | −1.989 *** (−2.91) | −1.883 * (−1.91) |
| Eastern region | 0.110 (0.86) | 0.148 (1.49) | 0.216 (1.52) | 0.255 (1.05) | 0.239 (0.88) |
| Market environment * Gender culture | | | | | −0.211 (−0.15) |
| Constant term | −1.915 *** (−5.06) | −1.408 *** (−17.65) | −1.832 (−1.05) | 0.937 (0.34) | 0.525 (0.13) |
| Chi$^2$ | 93.66 *** | 7.96 * | 53.02 *** | 115.15 *** | 115.23 *** |
| Pseudo R$^2$ | 0.1295 | 0.0048 | 0.0312 | 0.1580 | 0.1580 |
| Log-pseudolikelihood | −372.86308 | −834.93147 | −849.92897 | −337.26453 | −337.25278 |
| N | 1534 | 2464 | 2559 | 1465 | 1465 |

Note: The values in parentheses are *t* values; *, **, and *** sub-tables represent significance at 10%, 5%, and 1% levels.

## 6. Discussion and Conclusions

In recent years, along with the increase of female senior managers, factors influencing their employment have gradually become a focus of research. However, the existing literature has laid more emphasis on the number and proportion of female senior managers. Few studies have discussed the quality of employment. Therefore, a convincing explanation cannot be found in the present research. The key for explaining the phenomenon that women are shattering the glass ceiling and entering top management is the enterprise characteristics and environment. Thus, starting from the environment in which a company is involved, this paper combines the micro, meso, and macro levels and integrates them into a unified framework for analysis. By utilizing and matching the data of the World Bank Questionnaire Survey of Chinese Manufacturing Enterprises 2012, Chinese General Social Survey 2012, and relevant statistical yearbook data, this paper analyzes the influence of the corporate-level micro-environment, industry-level meso-environment, and regional-level macro-environment on the quality of employment of female senior managers and the influence mechanism. The main contribution of this paper is as follows. (1) It expands the theory of human resources and career acquisition from the perspective of the environment. (2) It gives attention to the influential factors of the quality of employment of female senior managers. (3) It provides a complete environment perspective that combines the micro-environment, meso-environment, and macro-environment. (4) It analyzes the mechanism of each environment's influence on the quality of employment of female senior managers. Fourth, this paper adopts Chinese samples to compensate for the existing inadequate literature in China, while providing valuable revelations for enterprise management and practice.

### 6.1. Conclusions

First, it outlines the profiles of companies with high CEO gender ratios and describes the general characteristics of such companies. In terms of the micro-environment, the enterprise size is small, with four to 17 employees. It is relatively new and has been established within the last six years. It is a subsidiary company and belongs to the consumer goods industry. It has female owners and has 80–90% of foreign shareholding and no state-owned shareholding. The product sales are mainly focused on the international market. Its financial risk is high, and the proportion of its self-owned funds is within 35–60%. In terms of the meso-environment, the company's competitors are mainly unregistered enterprises, and the company is in a better market environment than the average. In terms of the macro-environment, the company is located in areas with good economic development levels, with the GDP per capita ranging between RMB 81,658–91,202. It is 135.6 to 174.5 km away from Beijing, Shanghai, Guangzhou, Chongqing, and other national central cities, which are located in municipalities directly under the central government, and in areas with cultures that display more gender equality.

Second, the micro-environment, meso-environment, and macro-environment may all influence the CEO's gender, yet the degree of influence varies. In terms of the micro-environment, enterprise size and state-owned shareholding have a significantly negative impact on female CEOs: the probability of female CEOs decreased by 0.0142% and 0.116% respectively for every one unit added to the enterprise size and state-owned shareholding. Product features, industry characteristics, and female ownership have a significantly positive impact on the CEO's gender. The probability of female CEOs in consumer goods companies and companies with female owners increased by 0.0414% and 0.155%, if compared with non-consumer industries and companies without female owners, while the probability of female CEOs increased by 0.0289% for every one unit increased in the scope of product sales. In terms of the meso-environment, the market environment has a significantly negative impact on the quality of women's employment, and the probability of female CEOs decreased by 0.0498% for every one unit that it increased in the market environment. In terms of the macro-environment, the radiation effect and gender culture have a significant impact on the quality of women's employment. The probability of female CEOs increased by 0.129% for every one-unit increase in the distance between central cities.

In terms of gender culture, the probability of female CEOs increased by 0.178% for every one unit added to the gender culture.

Third, this paper further analyzes the influence mechanism of the environment on the CEO's gender; that is, the positive moderation of the macro-environment on the influence of the meso-environment on the CEO's gender. As discovered in this paper, the impacts of the market environment on the CEO's gender vary in different models. As revealed by the research findings, the gender culture in the macro-environment has a positive moderating effect on the market environment in the meso-environment. The reason why the moderating effect is not significant may be that the current gender culture is unfair to women and inherent prejudices toward women are widespread.

*6.2. Theoretical Contribution*

Theoretically, the traditional career development theory and career development theory explain to a certain extent why women become senior executives, but can not explain why they can not become CEOs. The important reason is that they ignore the external environment and gender differences. When we incorporate gender into existing theories, we can clearly see that women's development faces many difficulties, and it is difficult to make a difference on their own. The Achilles' heel of traditional theories is that they only focus on men and ignore women, and thus cannot explain the differences in career development between men and women [49].

Therefore, human resource theory should recognize the importance of the environment and gender differences, attach great importance to gender stereotypes and gender discrimination in industry careers, and re-establish work ethics of gender equality in order to promote sustainable development.

*6.3. Implication and Suggestion*

First, the government needs to adopt a multi-pronged approach and take different measures at the micro, meso, and macro levels to provide a good environment for women's career development. The research in this essay finds that the company's micro-environment, industry meso-environment, and regional macro-environment will all affect the number of female CEOs. In addition, the macro-environment has a positive adjustment effect on the meso-environment. Therefore, in order to achieve gender balance and sustainable development, we need to constantly improve corporate governance mechanisms, optimize the market environment, establish an equal gender culture, and provide equal opportunities for women's development.

Second, an enterprise needs to establish an organizational environment based on gender cooperation. The modern organization is controlled by men, and the corporate culture is dominated by masculinity. Therefore, it is necessary to establish an organizational environment where partner protection and cooperation are the main content, and ethical care for women is stressed.

Third, from the perspective of women themselves, the essay provides a reference for women's career development. Women's career development often faces the "ceiling dilemma", so prudence is required while choosing an occupation. In this regard, the research findings of this paper have certain reference significance for women's employment. For example, cities with a culture that displays more gender equality, better economic levels, and higher levels may be selected. Regarding companies, it is advisable to choose those involved in the consumer goods industry, with female owners and a broader product market. At the same time, companies with smaller enterprise size, lower state-owned shareholding, higher financial risks, and closer distance to central cities are preferable.

*6.4. Research Deficiency*

In terms of research content, due to the limitations of the sample data, this paper failed to study the relationship between the environment and its interaction on the gender impact of CEOs. At the same time, the environment is constantly changing; this paper lacks the dynamic analysis of the environment. Therefore, in future studies, we can further study the interaction between environments and the impact of their dynamic changes on female CEOs.

**Author Contributions:** Conceptualization, W.C.; Data curation, S.H.; Funding acquisition, W.C. and S.H.; Methodology, S.H. and W.C.; Project administration, W.C. and J.C.; Visualization, S.H.; Writing—original draft, S.H., W.C., and Y.F.; Writing—review and editing, S.H. and J.C.

**Funding:** This research was funded by The Philosophy and Social Sciences outstanding innovation team construction project of Jiangsu Province (No. 2015ZSTD006), The Development report of philosophy and social sciences of Ministry of education of China (No. 13JBG004), Postgraduate Research & Practice Innovation Program of Jiangsu Province (No. KYCX19_1072).

**Conflicts of Interest:** The authors declare no conflicts of interest.

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
