# Peer review of "Why Do Companies Choose Female CEOs?"

_sustainability, doi:10.3390/su11154070_

Reviewer 1 Report

report attached

Author Response

Dear reviewer: Hello! I am very glad to hear from you. Thank you very much for your contribution to this article. In order to seriously study the opinions of the four reviewers, improve the quality of the article,I take a long time to modify the manuscript, and I am sorry for that. Regarding the questions you raised, we have made serious revisions. The article adopts the revision mode. You can see the specific modification process.The amendments are specified in the annex. Thanks very much! yours sincerely Author

Reviewer 2 Report

First, I would suggest formulating the purpose of the article very clear. Reading the article begins with the expectation that strategic issues of Quality of Women's Employment will be addressed, considering the environmental development trends. Finally, the study reveals only the effect of combinations of the factors of environment at micro, meso, and macro level on Quality of Women's Employment.

I have great doubts regarding the use of the concept of "Scenario" in the paper. Most often the concept of "Scenario" is used when seeking to discuss the possible actions or development of events in the future.  Meanwhile, the authors simply study the links between different environmental factors and Quality of Women's Employment. The article does not examine the trend of environmental development, so I do not see the features of the scenario and the validity of the use of this term.

 I would also suggest paying attention to the title of the paper: 1) "What Type of Environment Improves the ...."  Micro, meso, and macro- do you investigate which level improves Quality of Women's Employment? 2) "....Study from an Enterprise  Scenario Perspective"  -  Do you have in mind the perspective of enterprise development trend?  I have not noticed that the article discusses strategic issues of the development of enterprise (-s).

Is not clear what the authors have in mind talking about the Quality of Women’s Employment. This construct is not defined and justified in the theoretical part .

Insufficient justification of  factors at different environmental levels. The authors limit themselves to mentioning that some or other scientists have researched them, but there is no deeper discussion describing the linkage between different factors and  Quality of Women’s Employment.

The discussion section states the results of research without discussing them further.

Author Response

(The authors gave the same response as above.)

Reviewer 3 Report

The study is interesting. The author(s) should clearly identify the aim of this research, specify what the quality of employment means. It is no clear what the quality of employment is. Is it possible that a better quality of employment means also quality of work and job..? Which motivations to the study that drive the description of the aims?The aspects about sustainability are not really present. The literature review should be deeply revised. Is the theoretical lens more macro-micro economic or managerial? It is no clear. The study provides data and information about a certain situation described but it should link the research design to sustainability perspective. Links between the research design and aims and sustainability are lacking.The analysis is mainly descriptive about how the environment influences the quality of employment but the relationships should be better clarified and explained. It is appreciable the efforts of the authors in providing data and information.The author(s) focus on a Chinese case study. So, probably, the title should be modified. Why does Chinese case study help understanding the development of women employments with regards to international or global scenario? Where is the discussion? So, pay attention to deeply revising the structure of the paper.

Author Response

Dear reviewer:

Hello!

I am very glad to hear from you. Thank you very much for your contribution to this article. In order to seriously study the opinions of the four reviewers, improve the quality of the article,I take a long time to modify the manuscript, and I am sorry for that.

Regarding the questions you raised, we have made serious revisions. The article adopts the revision mode. You can see the specific modification process. The specific answers are as follows:

1. Purpose and intention of this paper

In accordance with your suggestions, we have carefully considered the main objectives and contents of this article. As you said, strategy is a very important issue. Therefore, according to your suggestion, we have revised it.

Firstly, this paper studies it from a strategic perspective. The employment of CEO is a typical strategic process. This means that the choice of CEO is influenced by the strategic environment. Therefore, our research focuses on the impact of micro, meso and macro strategic environment on CEO gender. Of course, this paper only studies the static strategic environment, lacking the analysis of dynamic strategic environment, which is the inadequacy of this study.

Second, the purpose and theme of the study.As far as the current situation of enterprises is concerned, the number of female CEOs in enterprises is increasing. Why do some enterprises choose female CEOs while others choose male CEOs? Therefore, the question that this article considers is: Why do some enterprises choose female CEO? The core point of this paper is that there is no difference in competence between men and women. CEO gender is affected by strategic environment, which is complex. Macro-strategic environment has a moderating effect on CEO gender. Therefore, this paper mainly explains why enterprises choose female CEO from the perspective of strategic environment.

Third, the title of the article.In the title section, we have adopted your suggestion. The word Scenario is really not very good, and considering your other suggestions, we will change the title of the paper to “Why Do Women Serve as CEOs in Some Companies?——The Impact of Strategic Environment on CEO Gender”

2. The relationship between women and sustainable development

Thank you for your advice. We did not take this into account before, so as you suggest, we elaborate in the literature review section. It mainly includes two points:

1Sex and Gender Socialization

Sex is an objective fact. With the development of productivity, gender culture was gradually formed. It is well known that sex is caused by physiological differences, which is an objective fact that cannot be changed and must be recognized. In the primitive society, the basic needs of existence formed a desire for reproduction. Due to the ignorance of reproductive science, based on the fact that women can complete childbearing, respect and worship for women was formed in the primitive society. With the improvement of productivity level and the gradual formation of the social division of labor, men gradually mastered the dominant position of production in production and life due to their innate advantages. After that, family and marriage patterns also underwent major changes, and this change was also gradually strengthened, thus forming the male-dominated gender culture [10].

Under this gender culture, people gradually formed their understandings and even prejudice against male and female. Therefore, gender is the inevitable outcome of the socialization process of sex. The essence of gender is social construction. Gender culture is a basic component of human society and culture, and it had a major impact on people's production and life [11]. Gender culture refers to values, ethics, knowledge experience, customs, system norms and other ideologies and their manifestations developed on the basis of social characteristics, social behaviors and social relations between males and females [11]. Its core is to divide human beings into males and females, assign them different roles' connotation, identify different cultural instructions, and standardize different behavioral logic and development paths. Thus, the essence of gender is social construction and it is influenced by gender culture [12].

2CEO Genders and Corporate Sustainability

Firstly, CEO genders affect business decisions and behaviors. It is believed in the executive laddering theory that organizational outcomes are reflections of the values and cognitive abilities of the executive team, and its core theory is that organizational behavior serves as the individual interpretation of the environment which the executive team has to face, while interpretation of these personalities is a function of the values and personalities of executives and their teams [13, 14]. Existing studies also show that heterogeneity of executive characteristics may affect the impact of corporate policies, risks and performance [15]. Gender differences serve as the most prominent difference between male and female CEOs. Differences in values, behaviors and decision-making between males and females are caused due to their different growth experiences, which further affect the operation and decision-making of enterprises.

Secondly, compared with male CEOs, female CEOs' decision-making is more conducive to the corporate sustainability. As far as the relationship between females and corporate sustainability is concerned, on the one hand, as females’ important component in corporate governance, the increase of females in the board of directors has enhanced the corporate value [16], and female CEO improves the corporate performance [3]. In companies under complex environments, when the proportion of female executives is high, it can indeed produce positive and significant returns [17] than other companies with fewer female executives. At the same time, female CEOs promote corporate technological innovation more than male CEOs [10]. On the other hand, as for the outside of the company, the increase in the number of female executives improves the quality of corporate information disclosure [18]. Thus, female CEOs are conducive to the improvement of corporate value, the promotion of enterprise innovation, and the realization of corporate sustainability.

In summary, female CEOs are an important driving force for corporate sustainability. The participation of female CEOs is more conducive to corporate sustainability. Therefore, it is necessary to give full play to the role of females in the corporate sustainability, and there is a need to develop a new sustainable governance model that effectively combines males and femalesin the corporate governance [19]. In this model, CEO is at the core, and thus CEO gender is of great significance in the corporate sustainability.

3. Why choose China in the context of globalization?

Provide effective reference experience for countries with economies in transition by use of Chinese data. From the perspective of sample selection, there are few existing studies onChina. The Chinese economy is in a critical period of transformation and development, and its market mechanism is not yet sound, thus providing a broad space for exploring new concepts, theories and insights.Combined with the economic background of China's transition period, this paper provides effective reference experience for countries with transition economy or countries with imperfect market mechanism with regard to the increase of proportion of female CEOs and exertion of female CEOs’ abilities, and also provides more empirical evidence for deepening the cognition of female CEOs.

At the same time, the editorial department provided comments from other reviewers. According to these suggestions, we have also revised them. Please review them.

1. In the introduction, we reintroduce the contribution of this article.

Thank you very much for your suggestion. You have made feasible comments from 4 aspects and it is very reliable.

The contributions of this paper are mainly from three aspects, including: (1) research design, solving the problem "What insights can you provide based on your finding?", "Do they push forward our understanding?", (2) The meaning and suggestion of this article This part mainly answers the question "Do you have anysuggestions to improve the current regulation or practice?", (3) Finally, using the example of China to provide reference and help to more developing countries, this part answers "What? Should we do with your research?"

The revised content is as follows

The main contributions of this paper comprise (1) In terms of research design, this paper argues that there are no differences between males and females in business management ability, relaxes the original research hypothesis, and provides a complete strategic environment analysis framework for analysis of female executives, especially female CEOs, from the strategic environment, and clarifies the impact mechanism between environments.It is believed in this paper that gender is the result of social construction and it is impacted by the gender culture. Thus, there are no differences in ability between males and females. Meanwhile, corporate characteristics and environment serve as the key factors to explain how women break through the ceiling and enter into the top management of enterprises. Therefore, this paper provides a complete strategic environment analysis framework, further clarifies the impact mechanism of different environments on CEO genders, and enriches existing researches.(2) This paper is conducive to a reasonable view, understanding and evaluation of female CEOs, thus providing a reference for women's career development. The results of this paper show that female CEOs are impacted by gender culture, industry differences, corporate characteristics and other aspects. Females’ assumption of the office of CEO is the result of strategic environment. Thus, there is need to treat female CEOs more fairly and rationally, so as to provide good development space for female executives' career development and also provide reference for women's career development.(3) Provide effective reference experience for countries with economies in transition by use of Chinese data. From the perspective of sample selection, there are few existing studies on China. The Chinese economy is in a critical period of transformation and development, and its market mechanism is not yet sound, thus providing a broad space for exploring new concepts, theories and insights [9]. Combined with the economic background of China's transition period, this paper provides effective reference experience for countries with transition economy or countries with imperfect market mechanism with regard to the increase of proportion of female CEOs and exertion of female CEOs’ abilities, and also provides more empirical evidence for deepening the cognition of female CEOs. 

2.On Endogeneity

Endogeneity is indeed a difficult and important issue to deal with. Thank you very much for your suggestions. Referring to your suggestions, in dealing with endogenous problems, our original idea is to choose enterprise size as a tool variable, but this method is not feasible. The concrete operation of the method is as follows. The result of the graph is stata. From the result, the variables are not significant, so the method is not feasible.

(1) Firm size as a tool variable——unsuccessful

Tool variables are an effective way to alleviate endogenous problems. Generally speaking, tool variables need to meet the requirements of relevance and exogenous. Referring to the existing research, enterprise size is an important indicator, which can reflect the level of enterprise development as a whole. Therefore, this paper chooses the mean of enterprise size of the same city and the same industry in 2005 World Bank China Enterprise Survey data as the tool variable of enterprise micro-environment. If it does not meet the requirements of the same City and the same industry, the mean of enterprise size of the same city will be used as a substitute. 

This tool variable is based on the following three considerations: (1) It satisfies the "correlation" condition of the tool variable. Enterprise development and the production of the same industry in the city have a certain origin. The average size of the City-enterprise level in 2005 not only reflects the development of the city and the industry in the past, but also affects the development of enterprises in the city in 2012. (2) It meets the requirement of the hypothesis of "exogenous" of instrumental variables. Because the average size of urban-enterprise level in 2005 has little influence on the gender of CEO in 2012. (3) Data-based accessibility. Up to now, the World Bank has conducted four rounds of enterprise-level surveys in China, namely, in 2001 (5 cities), 2003 (18 cities), 2005 (120 cities) and 2012 (25 cities). However, the two rounds of surveys in 2001 and 2003 are 5 and 18 cities respectively, which are difficult to match with the survey in 2012. The 25 cities surveyed by the World Bank in 2012 were all included in 120 cities surveyed in 2005. Therefore, this paper calculates the average size of enterprises in the same city and industry by using the data of the World Bank Enterprise Survey in 2005, and uses this mean as a tool variable.

However, the results are not significant.

Based on 2004 data

Prob > chi2 = 0.7007This means that the instrumental variables are not significant, and so are other results.

Based on 2003 data

Based on 2002 data

(2) Other methods

Referring to SMJ's papersStrategic Management Journal, this paper adds control variables related to both independent variables and dependent variables. The references are as follows:

Schijven, M.; Hitt, M. A., The vicarious wisdom of crowds: Toward a behavioral perspective on investor reactions to acquisition announcements. Strategic Management Journal 2012,33, (11), 1247-1268. 

Foss, N. J.; Frederiksen, L.; Rullani, F., Problemformulation and problemsolving in selforganized communities: How modes of communication shape project behaviors in the free opensource software community. Strategic Management Journal2016,37, (13), 2589-2610. 

In terms of endogenous problems, this paper tries every possible to minimize the impact of endogenous variables on research results. Firstly, meso-environment and macro-environment belong to exogenous factors.Generally speaking, behaviors of a company are unable to affect the development of the entire industry, and it is even less likely to directly affect the political level and economic and cultural level of this region. Thus, endogenous problems in the meso-environment and macro-environment are greatly weakened.However, in the micro-environment, there may be a two-way causal relationship between CEO genders and the micro-environment under which enterprises are located [43]. For example, along with the corporate development, the construction of enterprise system is more and more perfect, which is in turn conducive to females’ becoming CEOs. There may be various other reasons such as the internal mutual supervision mechanism within the company [44], CEO power [45], enterprise size [23], etc.In addition, endogenous problems may also be caused by missing variables, measurement errors and other reasons. Therefore, endogenous problems need to be alleviated in this paper.    

In this paper, the method of adding as many control variables as possible that affect both explanatory variables and dependent variables is adopted for the reasons as follows (1) This method is an effective way to alleviate endogeneity, and it is adopted in international top-level journals.In this paper, the research results of Schijven, M., Hitt, M. A [46] and Foss, N. J., Frederiksen, L., Rullani, F. [47] are used as reference, and the method of adding control variables is used to alleviate endogeneity. (2) In the aspect of variable design in this paper, in order to reflect the impact of the corporate micro-environment on CEO genders, it is not available or possible to choose a core variable, because no one variable can reflect the company's overall level.That is to say, a core variable can not reflect the overall picture of the micro-environment. Therefore, the selection of enterprise scale and corporate equity structure as core variables can not reflect the actual situation of the corporate micro-environment. In conclusion, this method is chosen to alleviate endogeneity in this paper.

The specific practice of using this method is to increase the control variables correlated to both dependent variable and independent variable. The endogeneity in this paper is caused by variables in the corporate micro-environment. For this reason, we have added more corporate micro-environments as the control variable. The variables and setting methods are specified as follows: (1) The degree of informationization[1], there are rapid developments in the degree of informationization in recent years. Thus, such variable reflects the degree of corporate informationization, and this variable is set as the proportion of computers used in the work. The higher the proportion is, the higher the degree of corporate informationization will become. (2) Cooperative R&D[2]is set as a dummy variable mainly reflected in the cooperative R&D situation of enterprises. The fact of cooperative R&D is assigned as 1, otherwise 0; (3) Technology acquisition[3]is set as a virtual variable mainly reflected in the situation of corporate technology acquisition. The fact of technology acquisition is assigned as 1, otherwise 0;(4) International certification[4]is set as a virtual variable, and enterprises are assigned as 1 if they have passed international certification, otherwise 0.

In Table 9, the empirical results of endogeneity test are reported. From the results, all the models are significant as a whole. In terms of specific variables, except that the market environment in model 14 is not significant, other variables are significant, and there are no significant changes in the market environment in model 16 and Model 17. Therefore, we can think that endogenous problems have been effectively alleviated.

Table 9: Empirical Analysis of Endogenous Tests

Model 13

Model 14

Model 15

Model 16

Model 17

Enterprise size

-0.162**

-0.156*

-0.157*

(-2.02)

(-1.76)

(-1.77)

Enterprise age

-0.0623

-0.00792

-0.00927

(-0.33)

(-0.04)

(-0.05)

Company attributes

0.119

0.106

0.106

(0.34)

(0.27)

(0.27)

Industry characteristics

0.600***

0.614***

0.619***

(2.91)

(2.78)

(2.79)

Female ownership

2.285***

2.322***

2.315***

(6.35)

(6.05)

(6.00)

Foreign shareholding

0.644

0.568

0.558

(1.40)

(1.04)

(1.01)

State-owned shareholding

-1.780**

-1.844**

-1.831**

(-2.09)

(-2.08)

(-2.06)

Product market

0.437**

0.416**

0.407**

(2.18)

(2.10)

(2.07)

Financial risk

-0.639

-0.637

-0.615

(-1.52)

(-1.31)

(-1.25)

Competition order

-0.0332

0.0140

0.00823

(-0.16)

(0.06)

(0.04)

Market environment

0.227

-0.862**

0.997

(1.14)

(-1.96)

(0.18)

Economic level

0.295

0.179

0.262

(0.79)

(0.42)

(0.50)

Radiation effect

-0.935*

-2.512***

-2.292**

(-1.91)

(-3.19)

(-2.46)

City level

0.154

0.168

0.149

(1.52)

(1.35)

(1.06)

Gender culture

-1.826**

-2.915***

-2.588*

(-2.15)

(-2.58)

(-1.84)

Market environment* Gender culture

-0.861

(-0.34)

Informationization

1.086**

1.194***

0.845**

0.710

0.719

(2.36)

(2.82)

(1.97)

(1.44)

(1.45)

Cooperative R&D

-0.0367

-0.210

-0.374

-0.140

-0.148

(-0.11)

(-0.67)

(-1.20)

(-0.38)

(-0.39)

Technology acquisition

0.0641

0.130

0.0458

-0.255

-0.253

(0.25)

(0.56)

(0.21)

(-0.93)

(-0.92)

International certification

-0.302

-0.379*

-0.479**

-0.268

-0.261

(-1.30)

(-1.72)

(-2.29)

(-1.06)

(-1.02)

Constant term

-3.995***

-2.679***

-1.842

0.855

-0.809

(-5.34)

(-10.17)

(-0.38)

(0.15)

(-0.10)

chi2

97.72

12.50*

33.56***

110.91***

111.62***

Pseudo R2

0.1359

0.0144

0.0359

0.1606

0.1608

Log-pseudolikelihood

-356.71718

-399.76577

-425.34368

-325.83632

-325.77685

Sample size

1495

1497

1571

1435

1435

Note: The values in parentheses are tvalues; *, **, *** sub-tables represent significance at 10%, 5%, and 1% levels.

3. Other issues

(1) Literature

We also realize that the literature is very helpful to improve the content of the article, so we cite all of them.

(2) Explanation of the results

The interpretation of the results is also an important step to improve the quality of the paper. We explain the empirical results, and the red part of the article is the explanation part.

(3) Adjustment of tables

According to your request, the title of our table has been modified, and the specific content of the table title has been added.

Thanks very much!

yours sincerely

Author

[1]According to the questionnaireCNo8, the question is like this,currently, what percent of this establishment ‟s workforce regularly use computers in their jobs?

[2]According to the questionnaireCNo5, the question is like this, in the last three years, did this establishment spend on research and development activities contracted with other companies? The answer options are YES and No. 

[3]According to the questionnaire E6, the question is like this,does this establishment at present use technology licensed from a foreign-owned company, excluding office software?

[4]According to the questionnaire B8, the question is like this, does this establishment have an internationally-recognized quality certification?

(INTERVIEWER: if there is need for clarification, some examples are: ISO 9000 or 14000, or HACCP)

Reviewer 4 Report

The authors have used secondary data in a manner that promotes the testing of an interesting problem related to women in quality leadership positions in publicly traded enterprises. The splitting of the analysis in the micro, meso, and macro levels provides a different look at this challenge of women taking leadership roles in the workplace. 

This work serves as a foundation for future work. Although women have been in the workplace for many years, globally, women have not had the opportunity to be in higher level positions. The expectation of starting at the ground floor in an organization and working into management requires time within a company. Sadly, this data does not include education level, time of working, and time at the company. Education and job-jumping have resulted in women, and men, entering into management roles where they do not have a long history that was required in the past. 

Odds ratios would be interesting a possibility for future research. The only recommendation I have for the paper is to add to the titles of the tables such that a reader who is looking at a table, such as Table 6, will have a better idea of what was evaluated and presented in the table. Just having the title, Main Empirical Outcomes, leaves the reader having to go back to the text to find the description. A recommendation would be Main Empirical Outcomes for Analysis at the Micro-, Mesa-, and Macro-levels for Table 6. For Table 7 a possible recommendation is Main Empirical Outcomes of the Moderating Role of Culture Environment. 

It is good that the chi2 and t-values are included to give the reader an idea of the statistical significance and the confidence intervals for the coefficients presented in the tables. 

The implications and suggestions are very relevant and realistic.

Author Response

Dear reviewer:

Hello!

I am very glad to hear from you. Thank you very much for your contribution to this article. In order to seriously study the opinions of the four reviewers, improve the quality of the article,I take a long time to modify the manuscript, and I am sorry for that.

Regarding the questions you raised, we have made serious revisions. The article adopts the revision mode. You can see the specific modification process. The specific answers are as follows:

1. Education and working hours

Thank you very much for your suggestion. We also think these variables are very important. However, these variables are related to female executives themselves, which is not in line with the focus of this study. The main purpose of this paper is to study the impact of external environment on CEO gender. Therefore, we did not use them.

At the same time, influenced by the design of the questionnaire, the data is not provided in the questionnaire, so we can not obtain the data.

2. Increase the title of the table

According to your request, the title of our table has been modified, and the specific content of the table title has been added.

At the same time, the editorial department provided comments from other reviewers. According to these suggestions, we have also revised them. Please review them.

1. Purpose and intention of this paper

In accordance with your suggestions, we have carefully considered the main objectives and contents of this article. As you said, strategy is a very important issue. Therefore, according to your suggestion, we have revised it.

Firstly, this paper studies it from a strategic perspective. The employment of CEO is a typical strategic process. This means that the choice of CEO is influenced by the strategic environment. Therefore, our research focuses on the impact of micro, meso and macro strategic environment on CEO gender. Of course, this paper only studies the static strategic environment, lacking the analysis of dynamic strategic environment, which is the inadequacy of this study.

Second, the purpose and theme of the study.As far as the current situation of enterprises is concerned, the number of female CEOs in enterprises is increasing. Why do some enterprises choose female CEOs while others choose male CEOs? Therefore, the question that this article considers is: Why do some enterprises choose female CEO? The core point of this paper is that there is no difference in competence between men and women. CEO gender is affected by strategic environment, which is complex. Macro-strategic environment has a moderating effect on CEO gender. Therefore, this paper mainly explains why enterprises choose female CEO from the perspective of strategic environment.

Third, the title of the article.In the title section, we have adopted your suggestion. The word Scenario is really not very good, and considering your other suggestions, we will change the title of the paper to “Why Do Women Serve as CEOs in Some Companies?——The Impact of Strategic Environment on CEO Gender”

2. The relationship between women and sustainable development

Thank you for your advice. We did not take this into account before, so as you suggest, we elaborate in the literature review section. It mainly includes two points:

1Sex and Gender Socialization

Sex is an objective fact. With the development of productivity, gender culture was gradually formed. It is well known that sex is caused by physiological differences, which is an objective fact that cannot be changed and must be recognized. In the primitive society, the basic needs of existence formed a desire for reproduction. Due to the ignorance of reproductive science, based on the fact that women can complete childbearing, respect and worship for women was formed in the primitive society. With the improvement of productivity level and the gradual formation of the social division of labor, men gradually mastered the dominant position of production in production and life due to their innate advantages. After that, family and marriage patterns also underwent major changes, and this change was also gradually strengthened, thus forming the male-dominated gender culture [10].

Under this gender culture, people gradually formed their understandings and even prejudice against male and female. Therefore, gender is the inevitable outcome of the socialization process of sex. The essence of gender is social construction. Gender culture is a basic component of human society and culture, and it had a major impact on people's production and life [11]. Gender culture refers to values, ethics, knowledge experience, customs, system norms and other ideologies and their manifestations developed on the basis of social characteristics, social behaviors and social relations between males and females [11]. Its core is to divide human beings into males and females, assign them different roles' connotation, identify different cultural instructions, and standardize different behavioral logic and development paths. Thus, the essence of gender is social construction and it is influenced by gender culture [12].

2CEO Genders and Corporate Sustainability

Firstly, CEO genders affect business decisions and behaviors. It is believed in the executive laddering theory that organizational outcomes are reflections of the values and cognitive abilities of the executive team, and its core theory is that organizational behavior serves as the individual interpretation of the environment which the executive team has to face, while interpretation of these personalities is a function of the values and personalities of executives and their teams [13, 14]. Existing studies also show that heterogeneity of executive characteristics may affect the impact of corporate policies, risks and performance [15]. Gender differences serve as the most prominent difference between male and female CEOs. Differences in values, behaviors and decision-making between males and females are caused due to their different growth experiences, which further affect the operation and decision-making of enterprises.

Secondly, compared with male CEOs, female CEOs' decision-making is more conducive to the corporate sustainability. As far as the relationship between females and corporate sustainability is concerned, on the one hand, as females’ important component in corporate governance, the increase of females in the board of directors has enhanced the corporate value [16], and female CEO improves the corporate performance [3]. In companies under complex environments, when the proportion of female executives is high, it can indeed produce positive and significant returns [17] than other companies with fewer female executives. At the same time, female CEOs promote corporate technological innovation more than male CEOs [10]. On the other hand, as for the outside of the company, the increase in the number of female executives improves the quality of corporate information disclosure [18]. Thus, female CEOs are conducive to the improvement of corporate value, the promotion of enterprise innovation, and the realization of corporate sustainability.

In summary, female CEOs are an important driving force for corporate sustainability. The participation of female CEOs is more conducive to corporate sustainability. Therefore, it is necessary to give full play to the role of females in the corporate sustainability, and there is a need to develop a new sustainable governance model that effectively combines males and femalesin the corporate governance [19]. In this model, CEO is at the core, and thus CEO gender is of great significance in the corporate sustainability.

3. Why choose China in the context of globalization?

Provide effective reference experience for countries with economies in transition by use of Chinese data. From the perspective of sample selection, there are few existing studies onChina. The Chinese economy is in a critical period of transformation and development, and its market mechanism is not yet sound, thus providing a broad space for exploring new concepts, theories and insights.Combined with the economic background of China's transition period, this paper provides effective reference experience for countries with transition economy or countries with imperfect market mechanism with regard to the increase of proportion of female CEOs and exertion of female CEOs’ abilities, and also provides more empirical evidence for deepening the cognition of female CEOs.

4. In the introduction, we reintroduce the contribution of this article.

Thank you very much for your suggestion. You have made feasible comments from 4 aspects and it is very reliable.

The contributions of this paper are mainly from three aspects, including: (1) research design, solving the problem "What insights can you provide based on your finding?", "Do they push forward our understanding?", (2) The meaning and suggestion of this article This part mainly answers the question "Do you have anysuggestions to improve the current regulation or practice?", (3) Finally, using the example of China to provide reference and help to more developing countries, this part answers "What? Should we do with your research?"

The revised content is as follows

The main contributions of this paper comprise (1) In terms of research design, this paper argues that there are no differences between males and females in business management ability, relaxes the original research hypothesis, and provides a complete strategic environment analysis framework for analysis of female executives, especially female CEOs, from the strategic environment, and clarifies the impact mechanism between environments.It is believed in this paper that gender is the result of social construction and it is impacted by the gender culture. Thus, there are no differences in ability between males and females. Meanwhile, corporate characteristics and environment serve as the key factors to explain how women break through the ceiling and enter into the top management of enterprises. Therefore, this paper provides a complete strategic environment analysis framework, further clarifies the impact mechanism of different environments on CEO genders, and enriches existing researches.(2) This paper is conducive to a reasonable view, understanding and evaluation of female CEOs, thus providing a reference for women's career development. The results of this paper show that female CEOs are impacted by gender culture, industry differences, corporate characteristics and other aspects. Females’ assumption of the office of CEO is the result of strategic environment. Thus, there is need to treat female CEOs more fairly and rationally, so as to provide good development space for female executives' career development and also provide reference for women's career development.(3) Provide effective reference experience for countries with economies in transition by use of Chinese data. From the perspective of sample selection, there are few existing studies on China. The Chinese economy is in a critical period of transformation and development, and its market mechanism is not yet sound, thus providing a broad space for exploring new concepts, theories and insights [9]. Combined with the economic background of China's transition period, this paper provides effective reference experience for countries with transition economy or countries with imperfect market mechanism with regard to the increase of proportion of female CEOs and exertion of female CEOs’ abilities, and also provides more empirical evidence for deepening the cognition of female CEOs. 

5.On Endogeneity

Endogeneity is indeed a difficult and important issue to deal with. Thank you very much for your suggestions. Referring to your suggestions, in dealing with endogenous problems, our original idea is to choose enterprise size as a tool variable, but this method is not feasible. The concrete operation of the method is as follows. The result of the graph is stata. From the result, the variables are not significant, so the method is not feasible.

(1) Firm size as a tool variable——unsuccessful

Tool variables are an effective way to alleviate endogenous problems. Generally speaking, tool variables need to meet the requirements of relevance and exogenous. Referring to the existing research, enterprise size is an important indicator, which can reflect the level of enterprise development as a whole. Therefore, this paper chooses the mean of enterprise size of the same city and the same industry in 2005 World Bank China Enterprise Survey data as the tool variable of enterprise micro-environment. If it does not meet the requirements of the same City and the same industry, the mean of enterprise size of the same city will be used as a substitute. 

This tool variable is based on the following three considerations: (1) It satisfies the "correlation" condition of the tool variable. Enterprise development and the production of the same industry in the city have a certain origin. The average size of the City-enterprise level in 2005 not only reflects the development of the city and the industry in the past, but also affects the development of enterprises in the city in 2012. (2) It meets the requirement of the hypothesis of "exogenous" of instrumental variables. Because the average size of urban-enterprise level in 2005 has little influence on the gender of CEO in 2012. (3) Data-based accessibility. Up to now, the World Bank has conducted four rounds of enterprise-level surveys in China, namely, in 2001 (5 cities), 2003 (18 cities), 2005 (120 cities) and 2012 (25 cities). However, the two rounds of surveys in 2001 and 2003 are 5 and 18 cities respectively, which are difficult to match with the survey in 2012. The 25 cities surveyed by the World Bank in 2012 were all included in 120 cities surveyed in 2005. Therefore, this paper calculates the average size of enterprises in the same city and industry by using the data of the World Bank Enterprise Survey in 2005, and uses this mean as a tool variable.

However, the results are not significant.

Based on 2004 data

Prob > chi2 = 0.7007This means that the instrumental variables are not significant, and so are other results.

Based on 2003 data

Based on 2002 data

(2) Other methods

Referring to SMJ's papersStrategic Management Journal, this paper adds control variables related to both independent variables and dependent variables. The references are as follows:

Schijven, M.; Hitt, M. A., The vicarious wisdom of crowds: Toward a behavioral perspective on investor reactions to acquisition announcements. Strategic Management Journal 2012,33, (11), 1247-1268. 

Foss, N. J.; Frederiksen, L.; Rullani, F., Problemformulation and problemsolving in selforganized communities: How modes of communication shape project behaviors in the free opensource software community. Strategic Management Journal 2016,37, (13), 2589-2610. 

In terms of endogenous problems, this paper tries every possible to minimize the impact of endogenous variables on research results. Firstly, meso-environment and macro-environment belong to exogenous factors.Generally speaking, behaviors of a company are unable to affect the development of the entire industry, and it is even less likely to directly affect the political level and economic and cultural level of this region. Thus, endogenous problems in the meso-environment and macro-environment are greatly weakened.However, in the micro-environment, there may be a two-way causal relationship between CEO genders and the micro-environment under which enterprises are located [43]. For example, along with the corporate development, the construction of enterprise system is more and more perfect, which is in turn conducive to females’ becoming CEOs. There may be various other reasons such as the internal mutual supervision mechanism within the company [44], CEO power [45], enterprise size [23], etc.In addition, endogenous problems may also be caused by missing variables, measurement errors and other reasons. Therefore, endogenous problems need to be alleviated in this paper.    

In this paper, the method of adding as many control variables as possible that affect both explanatory variables and dependent variables is adopted for the reasons as follows (1) This method is an effective way to alleviate endogeneity, and it is adopted in international top-level journals.In this paper, the research results of Schijven, M., Hitt, M. A [46] and Foss, N. J., Frederiksen, L., Rullani, F. [47] are used as reference, and the method of adding control variables is used to alleviate endogeneity. (2) In the aspect of variable design in this paper, in order to reflect the impact of the corporate micro-environment on CEO genders, it is not available or possible to choose a core variable, because no one variable can reflect the company's overall level.That is to say, a core variable can not reflect the overall picture of the micro-environment. Therefore, the selection of enterprise scale and corporate equity structure as core variables can not reflect the actual situation of the corporate micro-environment. In conclusion, this method is chosen to alleviate endogeneity in this paper.

The specific practice of using this method is to increase the control variables correlated to both dependent variable and independent variable. The endogeneity in this paper is caused by variables in the corporate micro-environment. For this reason, we have added more corporate micro-environments as the control variable. The variables and setting methods are specified as follows: (1) The degree of informationization[1], there are rapid developments in the degree of informationization in recent years. Thus, such variable reflects the degree of corporate informationization, and this variable is set as the proportion of computers used in the work. The higher the proportion is, the higher the degree of corporate informationization will become. (2) Cooperative R&D[2]is set as a dummy variable mainly reflected in the cooperative R&D situation of enterprises. The fact of cooperative R&D is assigned as 1, otherwise 0; (3) Technology acquisition[3]is set as a virtual variable mainly reflected in the situation of corporate technology acquisition. The fact of technology acquisition is assigned as 1, otherwise 0;(4) International certification[4]is set as a virtual variable, and enterprises are assigned as 1 if they have passed international certification, otherwise 0.

In Table 9, the empirical results of endogeneity test are reported. From the results, all the models are significant as a whole. In terms of specific variables, except that the market environment in model 14 is not significant, other variables are significant, and there are no significant changes in the market environment in model 16 and Model 17. Therefore, we can think that endogenous problems have been effectively alleviated.

Table 9: Empirical Analysis of Endogenous Tests

Model 13

Model 14

Model 15

Model 16

Model 17

Enterprise size

-0.162**

-0.156*

-0.157*

(-2.02)

(-1.76)

(-1.77)

Enterprise age

-0.0623

-0.00792

-0.00927

(-0.33)

(-0.04)

(-0.05)

Company attributes

0.119

0.106

0.106

(0.34)

(0.27)

(0.27)

Industry characteristics

0.600***

0.614***

0.619***

(2.91)

(2.78)

(2.79)

Female ownership

2.285***

2.322***

2.315***

(6.35)

(6.05)

(6.00)

Foreign shareholding

0.644

0.568

0.558

(1.40)

(1.04)

(1.01)

State-owned shareholding

-1.780**

-1.844**

-1.831**

(-2.09)

(-2.08)

(-2.06)

Product market

0.437**

0.416**

0.407**

(2.18)

(2.10)

(2.07)

Financial risk

-0.639

-0.637

-0.615

(-1.52)

(-1.31)

(-1.25)

Competition order

-0.0332

0.0140

0.00823

(-0.16)

(0.06)

(0.04)

Market environment

0.227

-0.862**

0.997

(1.14)

(-1.96)

(0.18)

Economic level

0.295

0.179

0.262

(0.79)

(0.42)

(0.50)

Radiation effect

-0.935*

-2.512***

-2.292**

(-1.91)

(-3.19)

(-2.46)

City level

0.154

0.168

0.149

(1.52)

(1.35)

(1.06)

Gender culture

-1.826**

-2.915***

-2.588*

(-2.15)

(-2.58)

(-1.84)

Market environment* Gender culture

-0.861

(-0.34)

Informationization

1.086**

1.194***

0.845**

0.710

0.719

(2.36)

(2.82)

(1.97)

(1.44)

(1.45)

Cooperative R&D

-0.0367

-0.210

-0.374

-0.140

-0.148

(-0.11)

(-0.67)

(-1.20)

(-0.38)

(-0.39)

Technology acquisition

0.0641

0.130

0.0458

-0.255

-0.253

(0.25)

(0.56)

(0.21)

(-0.93)

(-0.92)

International certification

-0.302

-0.379*

-0.479**

-0.268

-0.261

(-1.30)

(-1.72)

(-2.29)

(-1.06)

(-1.02)

Constant term

-3.995***

-2.679***

-1.842

0.855

-0.809

(-5.34)

(-10.17)

(-0.38)

(0.15)

(-0.10)

chi2

97.72

12.50*

33.56***

110.91***

111.62***

Pseudo R2

0.1359

0.0144

0.0359

0.1606

0.1608

Log-pseudolikelihood

-356.71718

-399.76577

-425.34368

-325.83632

-325.77685

Sample size

1495

1497

1571

1435

1435

Note: The values in parentheses are tvalues; *, **, *** sub-tables represent significance at 10%, 5%, and 1% levels.

6. Other issues

(1) Literature

We also realize that the literature is very helpful to improve the content of the article, so we cite all of them.

(2) Explanation of the results

The interpretation of the results is also an important step to improve the quality of the paper. We explain the empirical results, and the red part of the article is the explanation part.

Thanks very much!

yours sincerely

Author

[1]According to the questionnaireCNo8, the question is like this,currently, what percent of this establishment ‟s workforce regularly use computers in their jobs?

[2]According to the questionnaireCNo5, the question is like this, in the last three years, did this establishment spend on research and development activities contracted with other companies? The answer options are YES and No. 

[3]According to the questionnaire E6, the question is like this,does this establishment at present use technology licensed from a foreign-owned company, excluding office software?

[4]According to the questionnaire B8, the question is like this, does this establishment have an internationally-recognized quality certification?

(INTERVIEWER: if there is need for clarification, some examples are: ISO 9000 or 14000, or HACCP)

Round  2

Reviewer 1 Report

Much improved. May add CSR (corporate social responsibility) aspect of female CEO. See Hong, B., Li, F., Minor, D., 2016, Corporate Governance and Executive Compensation for Corporate Social Responsibility, Journal of Business Ethics, 136 (1): 199-213 and Ikram A., Li, F. and Minor, D., 2019. CSR-Contingent Executive Compensation Contracts. Journal of Banking & Finance forthcoming. https://papers.ssrn.com/sol3/papers.cfm?abstract_id=3019985. I understand that you may not have the data, but you can attempt to discuss this important incentive in your context.

Author Response

Dear reviewers:

Hello!

I'm glad to receive your letter. Your suggestion is very important. We have revised it in response to your questions.

Point1May add CSR (corporate social responsibility) aspect of female CEO. See Hong, B., Li, F., Minor, D., 2016, Corporate Governance and Executive Compensation for Corporate Social Responsibility, Journal of Business Ethics, 136 (1): 199-213 and Ikram A., Li, F. and Minor, D., 2019. CSR-Contingent Executive Compensation Contracts. Journal of Banking & Finance forthcoming. https://papers.ssrn.com/sol3/papers.cfm?abstract_id=3019985.I understand that you may not have the data, but you can attempt to discuss this important incentive in your context.

Response 1: We have read your paper carefully. We think it is very important to enrich the content of this paper.One of the social responsibilities that companies should undertake is to actively protect women's legitimate rights and interests. Therefore, we quote in this article, see citations 42 and 43.

Finally, thank you again for your hard work.

Yours sincerely!

Authors

Reviewer 2 Report

Thank you for your efforts to improve the article. Nevertheless, the corrections raised new issues.

I have some doubts about the new title of the article "Why Do Women Serve as CEOs in Some Companies? The Impact of Strategic Environment is CEO Gender." I mean several aspects:

1)     What is a strategic environment? This term is used only in the title and in the introduction. Moreover, the introduction deals with the corporate strategic environment. Later you use macro, meso, micro environment terms. It is the same or different construct? As I was mentioned in the first review, I found out that the study reveals only the effect of combinations of the factors of environment at micro, meso, and macro level on...... (the next problem, see a remark No2). I would suggest just arguing why the different combinations of the factors of environment at micro, meso, and macro level impact on...... in the introduction.

2)     "The Impact of Strategic Environment on CEO Gender". The environment does not impact on CEO Gender. As you wrote in the line 202: "Sex is an objective fact". May be some combinations of the factors of environment require the characteristics that are more common for men or women?

3)      "Why Do Women Serve as CEOs in Some Companies? I think this part of the title is discriminatory and violates one of the principles of sustainable development (UN Global compact movement): Elimination of discrimination in respect of employment at an enterprise.

Pages 269-270: The same picture is presented twice

Although many changes have been made, there is no clarity what scientific problem is being addressed.

I'm still missing a justification of the factors at different environmental levels. The authors limit themselves to mentioning that some or other scientists have researched them, but there is no deeper discussion describing the linkage between different factors CEO gender.

The discussion section states the results of research without discussing them further.

Author Response

Dear reviewers:

Hello!

It's a pleasure to receive your letter. Thank you for your hard work. We have read your comments carefully,at least four times. Your suggestions are very helpful to improve the quality of the paper. Thank you very much.As we all know, theory is very important. We have also learned about the theory of women's career development through your suggestions, to make the logical relationship of the article clearer. Thank you very much for your suggestions. Your suggestions are very helpful to enrich the content of the article. Your suggestions are also the driving force for our continuous progress.

In response to your questions, we discussed and revised them in detail. The main contents are as follows.

Point1:  What is a strategic environment? This term is used only in the title and in the introduction. Moreover, the introduction deals with the corporate strategic environment. Later you use macro, meso, micro environment terms. It is the same or different construct? As I was mentioned in the first review, I found out that the study reveals only the effect of combinations of the factors of environment at micro, meso, and macro level on...... (the next problem, see a remark No2). I would suggest just arguing why the different combinations of the factors of environment at micro, meso, and macro level impact on...... in the introduction.

Respond1: Thank you very much for your suggestion. Your suggestion is very helpful to us. Your proposal includes two questions.

Question 1: What is the strategic environment and what is its classification?

In response to your suggestion, we have mainly adopted three modifications, the content of which is 2.3.1. The logical structure of this part is: (1) Strategic environment is very important for business operation; (2) Employment of CEO is an important strategic decision-making, which needs to build a complete strategic environment; (3) Definition and classification of strategic environment. The above modifications are mainly in 2.3.1.

Question 2: The regulatory mechanism of macro-environment to meso-environment.

We have read your suggestion carefully. We think it is very helpful. Thank you. To this suggestion, we shall take some measures:

Firstly, we summarize the impact of various factors in the strategic environment on CEO gender. As shown in Figure 3, we also make a detailed analysis of the impact of the market environment, which is shown in 5.1.2.

Secondly, in view of the impact mechanism of macro-environment on meso-environment, we elaborate on it in 5.2. The impact mechanism is shown in Figure 4. We believe that there are three main roles: (1) to respect and recognize women's physiological differences; (2) to recognize women's social values; (3) to help women's socialization and exert female talents.

Point2:   "The Impact of Strategic Environment on CEO Gender". The environment does not impact on CEO Gender. As you wrote in the line 202: "Sex is an objective fact". May be some combinations of the factors of environment require the characteristics that are more common for men or women?

Respond2: Thank you for your suggestion. As mentioned earlier, we further discuss the impact of the strategic environment on CEO gender, especially the role of gender culture. The reason why women become CEOs is because they adapt to the changes of the strategic environment, and the characteristics of masculinity are the result of women's socialization. Therefore, this article does not conflict with your suggestion. Maybe we did not write clearly, so we also carried out Revised.

Point3:  "Why Do Women Serve as CEOs in Some Companies? I think this part of the title is discriminatory and violates one of the principles of sustainable development (UN Global compact movement): Elimination of discrimination in respect of employment at an enterprise.

Respond3: Thank you very much for your advice, this is a very important suggestion! And it could easily be overlooked! The elimination of discrimination in employment is indeed an important principle for achieving sustainable development, which is the starting point of this paper. However, the current study, researchers mainly from the perspective of a single study, lack of integration perspective, and an integrated, systematic perspective better find the problem. So, considering your suggestion, and the research purposes and issues in this article, we change the title to "Why do companies choose female CEOs? - The Research based on Strategic Environment"

Point4: Pages 269-270: The same picture is presented twice

Respond4: Thank you for your suggestion. Maybe because of the software problem, we didn't find this problem. We will upload the PDF file, thank you.

Point5: Although many changes have been made, there is no clarity what scientific problem is being addressed.

Respond5: Thank you for your suggestion. It is important to clarify the research issues of the paper. The structure of this paper is:

 The theoretical background of this paper is: the theory of occupation status acquisition believes that occupational status depends mainly on personal achievement, including education level and effort level. At the same time, human capital theory believes that education level and work experience have a great relationship with personal career development.

The reality of this article is: In recent years, women's education level and social participation have been mentioned, but women still do not enter the company's top management, and few CEOs are women.

This forms a gap between theory and reality. Both the occupational status acquisition theory and the human capital theory emphasize the importance of personal factors to career development, and the female individual factors have been improved, but the proportion of women in the CEO is still low.

Therefore, this paper believes that the existing theory overemphasizes its own factors, does not conduct research from the perspective of strategic environment and gender, and needs to supplement the perspective of the external environment. So this article explains from an integrated perspective why companies choose female CEOs.

Point6: I'm still missing a justification of the factors at different environmental levels. The authors limit themselves to mentioning that some or other scientists have researched them, but there is no deeper discussion describing the linkage between different factors CEO gender.

Respond6:

thank you very much for your suggestion. This part of the content has been replied in Respond1. First, we summarize the impact of various factors in the strategic environment on CEO gender, as shown in Figure 3. We also analyzed the impact of the market environment in detail, as described in 5.1.2.

Secondly, the mechanism of the impact of the macro environment on the mesoscopic environment is described in detail in 5.2. The impact mechanism is shown in Figure 4. We believe that there are three main functions, (1) to respect and recognize the physiological differences of women; (2) to recognize the social value of women; (3) to promote the socialization of women and to develop female talents.

Point7: The discussion section states the results of research without discussing them further.

Respond7: Thank you for your suggestion. Your suggestion is very helpful for improving the depth of the paper. Thank you.

First, we emphasize the theoretical contribution of this paper and reflect on traditional theory, such as 6.2. The current theory cannot explain why women can't become CEOs. The important reasons ignore the strategic environment and ignore the gender differences. This paper has studied from the perspective of strategic environment and expanded the existing theory.

Second, we revised our policy recommendations to make it clearer, such as 6.3. Mainly from the government, companies, women themselves made recommendations.

Finally, thank you again for your hard work.

Yours sincerely!

Authors

Reviewer 3 Report

The efforts of author(s) in improving their manuscript are very appreciable. In re-designing research setting and development, it is still necessary to focus on aims, purposes and research questions that seem to be not yet properly clear and adequately positioned. The new title is better than previous one but it seems to look at another perspective on which the author(s) focus on. The section about the relationship between the topic and sustainability also in terms of corporate sustainability is not yet fully convincing. Better defining the concept of strategic environment in the literature. Better refining the concept of corporate sustainability and better linking with the discourse the author(s) tend to follow. The theoretical background and lenses should be more focused and specified, coherent and adequate to reconstruction of research design.

Author Response

Dear reviewers:

Hello!

It's a pleasure to receive your letter. Thank you for your hard work. We have read your comments carefullyat least four times. Your suggestions are very helpful to improve the quality of the paper. Thank you very much.

As we all know, theory is very important. We have also learned about the theory of women's career development through your suggestions, to make the logical relationship of the article clearer. Thank you very much for your suggestions. Your suggestions are very helpful to enrich the content of the article. Your suggestions are also the driving force for our continuous progress.

In response to your questions, we discussed and revised them in detail. The main contents are as follows.

Point1:The efforts of author(s) in improving their manuscript are very appreciable. In re-designing research setting and development, it is still necessary to focus on aims, purposes and research questions that seem to be not yet properly clear and adequately positioned. The theoretical background and lenses should be more focused and specified, coherent and adequate to reconstruction of research design.

Respond1: Thank you for your suggestion. Based on your suggestions, we reorganized the paper from the theoretical background, the reality and phenomena, research issues and priorities, problem analysis, theoretical contributions, policy recommendations, etc., making this article logical. The revised content of this article is:

1. Theoretical background

the theory of occupation status acquisition believes that occupational status depends mainly on personal achievement, including education level and effort level. At the same time, human capital theory believes that education level and work experience have a great relationship with personal career development.

2.the reality and phenomena

In recent years, women's education level and social participation have been mentioned, but women still do not enter the company's top management, and few CEOs are women.

3. Research issues and priorities

This forms a gap between theory and reality. Both the occupational status acquisition theory and the human capital theory emphasize the importance of personal factors to career development, and the female individual factors have been improved, but the proportion of women in the CEO is still low.

Therefore, this paper believes that the existing theory overemphasizes its own factors, does not conduct research from the perspective of strategic environment and gender, and needs to supplement the perspective of the external environment. So this article explains from an integrated perspective why companies choose female CEOs.

4. problem analysis

First, we summarize the impact of various factors in the strategic environment on CEO gender, as shown in Figure 3. We also analyzed the impact of the market environment in detail, as described in 5.1.2.

Secondly, the mechanism of the impact of the macro environment on the mesoscopic environment is described in detail in 5.2. The impact mechanism is shown in Figure 4. We believe that there are three main functions, (1) to respect and recognize the physiological differences of women; (2) to recognize the social value of women; (3) to promote the socialization of women and to develop female talents.

5. theoretical contributions

we emphasize the theoretical contribution of this paper and reflect on traditional theory, such as 6.2. The current theory cannot explain why women can't become CEOs. The important reasons ignore the strategic environment and ignore the gender differences. This paper has studied from the perspective of strategic environment and expanded the existing theory.

6. policy recommendations

we revised our policy recommendations to make it clearer, such as 6.3. Mainly from the government, companies, women themselves made recommendations.

In summary, we have reorganized the research from the above six aspects, making the logic of the article clearer, thank you very much!

Point2:The new title is better than previous one but it seems to look at another perspective on which the author(s) focus on. The section about the relationship between the topic and sustainability also in terms of corporate sustainability is not yet fully convincing. Better refining the concept of corporate sustainability and better linking with the discourse the author(s) tend to follow.

Respond2: thank you very much for your suggestion. In response to your suggestion, we have reworked the 2.2 part of this article, jumping out of the corporate framework, and demonstrating the importance of women's sustainable development from the perspective of social development. The main structure is as follows:

First of all, from the perspective of motivation, sustainable development requires the participation of the whole society, and women are an important force for achieving sustainable development.

Secondly, from the perspective of gender equality, gender equality is an inevitable choice for achieving sustainable development. In September 2015, the United Nations Sustainable Development Summit was held. The meeting formally approved the “Change Our World – 2030 Agenda for Sustainable Development” and put forward the goal of “achieving gender equality and empowering all women and girls”.

Finally, in terms of processes and abilities, women have the ability to promote sustainable development.

Point3:Better defining the concept of strategic environment in the literature.

Respond2:In response to your suggestion, we have mainly adopted three modifications, the content of which is 2.3.1. The logical structure of this part is: (1) Strategic environment is very important for business operation; (2) Employment of CEO is an important strategic decision-making, which needs to build a complete strategic environment; (3) Definition and classification of strategic environment. The above modifications are mainly in 2.3.1.

Finally, thank you again for your hard work.

Yours sincerely!

Authors

Round  3

Reviewer 2 Report

Thank you for the responses and the effort by improving the paper. Nevertheless, I have some remarks. To be honest, I still did not understand the definition of strategic environment. I will ask a very simple question. What is the difference between a strategic environment and an environment? Figure 1 shows the Connotation and Classification of Environment. What we should change in Figure 1 to be a strategic environment. I think we are starting to create terms, but we do not disclose its content. moreover, I have spent a lot of time looking for the interpretation of this term i. Unfortunately, I didn’t find that the term is used in management science literature, but it is devoted to military and political science. However, if you think otherwise, just answer my question regarding the differences between the environment and the strategic environment.

Regarding the title. I think the first sentence is enough: Why do companies choose female CEOs ? Maybe it is simple but clear and reflects the main idea of the article.

Author Response

Dear reviewers:

I'm glad to receive your letter. Thank you for your hard work.Your suggestion is very important. We have revised it according to your advises.

Point1:To be honest, I still did not understand the definition of strategic environment.I will ask a very simple question. What is the difference between a strategic environment and an environment? Figure 1 shows the Connotation and Classification of Environment. What we should change in Figure 1 to be a strategic environment. I think we are starting to create terms, but we do not disclose its content. moreover, I have spent a lot of time looking for the interpretation of this term i. Unfortunately, I didn’t find that the term is used in management science literature, but it is devoted to military and political science. However, if you think otherwise, just answer my question regarding the differences between the environment and the strategic environment.

Response1:Frankly speaking, a clear concept is a very important aspect of the paper. So, your suggestion is very important. Thank you very much for your suggestion.

The original idea of this article is that hiring a CEO is a strategic behavior because the CEO's tenure is longer and has the most extensive impact on the company. Therefore, we believe that hiring a CEO is influenced by the strategic environment.Of course, this is a view from a strategic management perspective.However, as you said, there is no essential difference between the strategic environment and the environment in this article, so we decided to adopt your suggestion. Therefore, we modify the strategic environment to the environment.

Point2:Regarding the title. I think the first sentence is enough: Why do companies choose female CEOs ? Maybe it is simple but clear and reflects the main idea of the article.

Response2: Thank you very much for your suggestion. The title is very important to the article, so we follow your advice.

Finally, thank you again for your hard work.

Yours sincerely!

Authors

Reviewer 3 Report

The efforts in modifying and improving the manuscript are appreciable. Some suggestions were accepted and implemented. Some replies to point 2 are not fully convincing and could better explained and presented. Revisiting the paper in order to make it more cohesive and integrated in all the parts.

Author Response

Dear reviewers:

I'm glad to receive your letter. Thank you for your hard work.Your suggestion is very important. We have revised it according to your advises.

Point1:The efforts in modifying and improving the manuscript are appreciable. Some suggestions were accepted and implemented. Some replies to point 2 are not fully convincing and could better explained and presented. Revisiting the paper in order to make it more cohesive and integrated in all the parts.

Response1:Your suggestion is very important. Thank you very much for your suggestion.

This part of the revision is in section 2.2 of the article.

First, we discuss the relationship between women and sustainable development.We found that they interacted with each other, women promoted sustainable development, and sustainable development supported women's development. Therefore, we use Figure 1 to show the relationship between them.

Secondly, as shown in Figure 2,we further discuss the mechanism for women to promote sustainable development, mainly from three aspects: political, economic and social.

Finally, thank you again for your hard work.

Yours sincerely!

Authors
